# NETWORK SHAPE AUTOMATA: A BRAIN NETWORK INSPIRED COLLABORATIVE FILTER FOR LINK PREDICTION IN BIPARTITE COMPLEX NETWORKS AND RECOMMENDATION SYSTEMS

## ABSTRACT

In recommendation systems, representing user-item interactions as a bipartite network is a fundamental approach that provides a structured way to model relationships between users and items, allowing for efficient predictions via network science. Collaborative filtering is one of the most widely used and actively researched techniques for recommendation systems, its rationale is to predict user preferences based on shared patterns in user interactions, and vice versa. Memory-based collaborative filtering relies on directly analyzing user-item interactions to provide recommendations using similarity measures, and differs from model-based collaborative filtering which builds a predictive model using machine learning techniques such as neural networks. With the rise of machine learning, memory-based collaborative filtering has often been overshadowed by model-based approaches. However, the recent success of SSCF, a newly proposed memory-based method, has renewed interest in the potential of memory-based approaches. In this paper, we propose Network Shape Automata (NSA), a memory-based collaborative filtering method grounded in the connectivity shape of the bipartite network topology. NSA leverages the Cannistraci-Hebb theory proposed in network science to define brain-inspired network automata, using this paradigm as the foundation for its similarity measure. We evaluate NSA against a range of advanced collaborative filtering methods, both memory-based and model-based, across 13 middle-scale bipartite network datasets spanning complex systems domains such as social networks and biological networks and 3 classical large-scale recommendation datasets including Gowalla, Yelp2018, Amazon-book. Results show that NSA consistently achieves strong performance across diverse datasets and evaluation metrics, ranking most often first on average. Notably, NSA demonstrates strong robustness to network sparsity, while preserving the simplicity, interpretability, and training-free nature of memory-based methods. As a pioneering effort to bridge link prediction and recommendation tasks, NSA not only highlights the untapped potential of memory-based collaborative filtering but also demonstrates the effectiveness of the Cannistraci-Hebb theory in modeling network evolution within recommendation systems.

## 1 INTRODUCION

In many real-world scenarios, relationships between entities can be modeled as bipartite networks, where edges only exist between two disjoint sets of nodes, such as users and items (Watts & Strogatz, 1998; Albora et al., 2022; Tacchella et al., 2012; Straccamore et al., 2022). Predicting new links in these networks, often framed as recommendation, is a crucial task for improving user experience and system efficiency (Ricci et al., 2021). Collaborative filtering (CF) is one of the most widely used approaches in recommendation systems (Goldberg et al., 1992; Schafer et al., 2007), with memory-based and model-based methods as two major branches (Chen et al., 2018). While memory-based CF methods are simple and highly interpretable, they have long been considered less competitive in performance compared to more complex model-based methods.

However, recent advance, Sapling Similarity Collaborative Filtering (SSCF), has shown that memory-based approaches still hold significant promise (Albora et al., 2023). SSCF leverages a new similarity measure and achieves state-of-the-art performance on benchmark datasets, outperforming all the other models. This suggests that the full potential of memory-based methods has yet to be realized, particularly if better ways of capturing structural information in networks can be found.

Most traditional memory-based CF methods rely on basic node similarity measures, often limited to shared neighbors, which overlook deeper topological insights. In contrast, network science offers rich theoretical foundations for understanding link formation. The Cannistraci-Hebb (CH) theory (Muscoloni et al., 2018; 2020), inspired by brain connectivity, emphasizes the importance of local community structures (Cannistraci et al., 2013) rather than just node-level features. CH-based methods are network automata rules that have shown strong performance in various link prediction tasks and have even been used to sparsify neural networks while preserving accuracy (Abdelhamid et al., 2023; Zhang et al., 2023).

Table 1: **Number of real-world network datasets tested by different methods**. For NSA, considering the recommendation is made from both views of two sets of nodes, the number of datasets is multiplied by 2.

| Algorithm | Year | Networks | Ref |
|---|---|---|---|
| NGCF | 2019 | 3 | Wang et al. (2019) |
| LightGCN | 2020 | 3 | He et al. (2020) |
| UltraGCN | 2021 | 4 | Mao et al. (2021b) |
| SimpleX | 2021 | 11 | Mao et al. (2021a) |
| LT-OCF | 2021 | 3 | Choi et al. (2021) |
| BSPM | 2022 | 3 | Choi et al. (2023) |
| SSCF | 2023 | 5 | Albora et al. (2023) |
| XSimGCL | 2023 | 4 | Yu et al. (2023) |
| **NSA** | **2025** | **13 x 2** | **Ours** |

Motivated by the theoretical and empirical strength of CH theory, we propose Network Shape Automata (NSA), a novel memory-based collaborative filtering method that fully leverages network topology for recommendation. NSA adheres to the classical architecture of memory-based CF, yet redefines similarity computation based on local topological features derived from CH theory. We evaluate NSA on various benchmark datasets from both the recommendation and link prediction domains. Results demonstrate that NSA consistently achieves competitive, and in some cases superior, performance compared to state-of-the-art models, while preserving the simplicity and transparency of memory-based systems. Our work highlights the overlooked potential of structural information in network-based recommendations and presents NSA as a bridge between interpretable design and high recommendation accuracy.

Here, we present our main contribution in this work as follows:

**Network Shape Automata (NSA):** We propose NSA, a memory-based collaborative filtering method that integrates CH theory into similarity computation using local network topology.

**Comprehensive Evaluation:** NSA was evaluated on 13 datasets spanning recommendation and link prediction tasks which is more comprehensive than existing works as shown in Table 1, consistently showing stable and often superior performance. Extensive hyperparameter tuning (over 105,300 model assessments) ensured fair and reproducible comparisons.

**Large-Scale Dataset Robustness:** NSA maintains stable and strong performance on large datasets, including Gowalla, Yelp2018, and Amazon-Book, demonstrating its scalability and reliability.

**Leveraging Structural Information:** NSA effectively exploits network structural features to capture meaningful user-item relationships, maintaining high accuracy even with sparse interactions, while remaining interpretable.

## 2 RELATED WORK

### 2.1 BIPARTITE NETWORK PROJECTION

Bipartite networks consist of two disjoint sets of nodes with edges only between nodes of different sets (Zhou et al., 2007), and are commonly used to represent real-world relationships, for example, users and items in recommendation systems. In such applications, the two sets typically correspond to users and items, with edges representing interactions such as purchases, views, or ratings. Depending on the nature of these interactions, bipartite networks can be divided into two categories: non-unary rating, where links carry explicit preference scores; and unary rating, where links only indicate the presence or absence of interaction, without expressing degrees of preference (Goldberg et al., 1992).

This paper focuses on the unary rating scenario, where collaborative filtering methods are widely adopted.

Bipartite network projection, or one-mode projection, transforms a bipartite network into two monopartite ones by connecting nodes of the same type if they share common neighbors (Zhou et al., 2007). This process captures similarity within a single node set and serves as a compressed representation of the original bipartite structure. However, such compression inevitably loses some relational detail, making the choice of edge weighting in the projected network critical for preserving meaningful information (Fan et al., 2007; Newman, 2001). Different weighting methods emphasize different aspects of the original network and are chosen based on the analytical goals of the projection.

## 2.2 Collaborative Filtering

Recommendation systems are essential tools for delivering personalized content by predicting user preferences based on historical interactions. To address this task, various approaches have been proposed, including content-based approach (Brusilovski et al., 2007), collaborative filtering (Breese et al., 2013), and hybrid models that combine multiple strategies. Among them, collaborative filtering (CF) stands out for its effectiveness and broad adoption, relying on user behavior shared patterns rather than item attributes.

Collaborative filtering can be further classified into memory-based and model-based methods (Chen et al., 2018; Su & Khoshgoftaar, 2009).

Memory-based approaches predict user preferences by computing similarities between users or items, with various similarity measures developed to improve recommendation accuracy. The structure of different memory-based methods is largely the same, with the choice of similarity measure being the key differentiator. Widely used similarity measures in memory-based approaches include Common Neighbors (Liben-Nowell & Kleinberg, 2003), Jaccard (Jaccard, 1901), Resource Allocation Index (Zhou et al., 2009), Cosine Similarity, and Pearson Correlation Coefficient (Shardanand & Maes, 1995). Most of these measures estimate similarity based on the common neighbors of two nodes. Notably, the recently proposed Sapling Similarity Collaborative Filtering (SSCF) introduces a probabilistic perspective that enables negative similarity modeling, offering improved performance (Albora et al., 2023).

Model-based approaches, in contrast, learn predictive models from user-item interactions using machine learning techniques. Recent advancements focus on neural network methods, particularly Graph Convolutional Networks (GCNs), which capture high-order user-item connectivity (Chen et al., 2020). These include NGCF, an early and influential method that introduced graph-based message passing for collaborative filtering (Wang et al., 2019); LightGCN, which simplified this framework while achieving stronger performance (He et al., 2020); SimpleX, which further optimized the model design for efficiency (Mao et al., 2021a); UltraGCN, which avoided explicit graph convolution by modeling global interactions (Mao et al., 2021b); LT-OCF, which models user and item embedding evolution over continuous time using neural ODEs with learnable interaction timestamps, thereby effectively capturing temporal dynamics (Choi et al., 2021); and BSPM, which uses a blurring-sharpening process to perturb and refine interactions, and is regarded as a diffusion-based approach rather than a conventional neural network method (Choi et al., 2023); and XSimGCL, which incorporates contrastive learning into graph-based recommendation in an extremely simple yet effective way by perturbing embeddings and enforcing consistency, thus significantly improving robustness and performance under sparse interactions (Yu et al., 2023). A very recent CL-based method, NLGCL (Xu et al., 2025), further pushes this direction by leverageing naturally contrastive views between neighbor layers within GNNs for contrastive learning.

## 2.3 Cannistraci-Hebb Theory

CH rules are network automata for estimating the likelihood of a non-observed link to appear in the network. These rules are classified as network automata because they utilize only local information to infer the score of a link in the network without need of pre-training of the rule. Note that CH rules are predictive network automata that differ from generative network automata which are rules created to generate artificial networks (Barabási & Albert, 1999; Papadopoulos et al., 2012; Muscoloni & Cannistraci, 2018). The concept of *network automata* was originally introduced by Wolfram (Wolfram & Gad-el Hak, 2003) and later formally defined by Smith et al. (Smith et al., 2011) as a general framework for modeling the evolution of network topology. Given an unweighted and undirected

adjacency matrix $X(t)$ at time $t$, in a network automaton the states of links evolve over time according to a rule that depends only on local topological properties computable from a portion of the adjacency matrix $\tilde{X}(t) \subset X(t)$:

$$\tilde{X}(t+1) = F(\tilde{X}(t)) \tag{1}$$

Network shape intelligence is an emerging paradigm that tries to perform link prediction by exploiting the intrinsic topological structure of real-world networks, without relying on training or external data. The core idea is to treat the network itself as both input and source of knowledge, enabling unsupervised predictions based solely on local connectivity patterns (Abdelhamid et al., 2023). A representative advancement in this area is the Cannistraci-Hebb (CH) theory, which extends Hebbian learning, originally proposed in neuroscience, to the domain of complex network analysis (Cannistraci, 2018).

Hebbian learning posits that coactivated neurons tend to form connections and was generalized into the Local-Community Paradigm (LCP) (Cannistraci et al., 2013). LCP assumes that new links are more likely to form within local communities, where nodes are densely connected and related. CH theory formalizes this through two structural tendencies: maximization of internal local community links (iLCL) and minimization of external local community links (eLCL) (Muscoloni et al., 2018; 2020). Based on these principles, different versions of CH indexes have been proposed that focus different properties of networks (CHn). In addition, multi-scale variants (Ln) are introduced to account for different community sizes, based on the path length between node pairs.

CH-based link predictors have shown strong empirical performance across different domains. In particular, Cannistraci et al. demonstrated that a CH-inspired predictor outperformed AlphaFold in protein-protein interaction prediction (Abdelhamid et al., 2023). Furthermore, neural networks with CH-based sparse connectivity, retaining only 1% of original links, achieved comparable or better results than fully connected models (Zhang et al., 2023), suggesting the potential of biologically-inspired, ultra-sparse architectures.

These insights underscore the predictive power of topology alone and provide theoretical support for applying CH theory to recommendation systems, especially in settings where data sparsity or lack of supervision poses significant challenges.

## 3 NETWORK SHAPE AUTOMATA

To formally present our approach, we begin by introducing the fundamental definitions. Consider a bipartite network representing the recommendation system, where the set of user nodes is denoted by $U$ and the set of item nodes by $\Gamma$. The cardinalities $|U|$ and $|\Gamma|$ indicate the total number of users and items, respectively. The network structure is encoded by an adjacency matrix $M \in \mathbb{R}^{|U| \times |\Gamma|}$, where each entry $M_{u\gamma} = 1$ if user $u$ is connected to item $\gamma$, and 0 otherwise. As we focus on unary rating scenarios, the network is assumed to be unweighted. The degree of a user node $u$ is defined as $d_u = \sum_{\gamma=1}^{|\Gamma|} M_{u\gamma}$, while the degree of an item node $\gamma$ is denoted by $d_\gamma$. The set of common neighbors between users $i$ and $j$ is denoted $CN_{ij}$, and similarly, the common neighbors between items $\alpha$ and $\beta$ are denoted $CN_{\alpha\beta}$.

Then, we introduce Network Shape Automata (NSA) which can be treated as memory-based collaborative filtering in a topological way. Specifically, NSA follows the steps described in the subsections 3.1 to 3.4 below, illustrated in Fig 1.

### 3.1 CH SCORING

As the core component of NSA, we calculate the similarity between different pairs of nodes based on CH theory.

**CH index** Inspired by CH theory (Muscoloni et al., 2018), the basis of the similarity is CH indexes, including CH3-L2 (Muscoloni et al., 2020) and CH3.1-L2 (Zhao et al., 2025). CH3-L2 is the version based on local community for path of length 2 and takes into account only the minimization of external links, of which the formula is

$$\text{CH3-L2(i, j)} = \sum_{k \in L2} \frac{1}{de_k + 1} \tag{2}$$

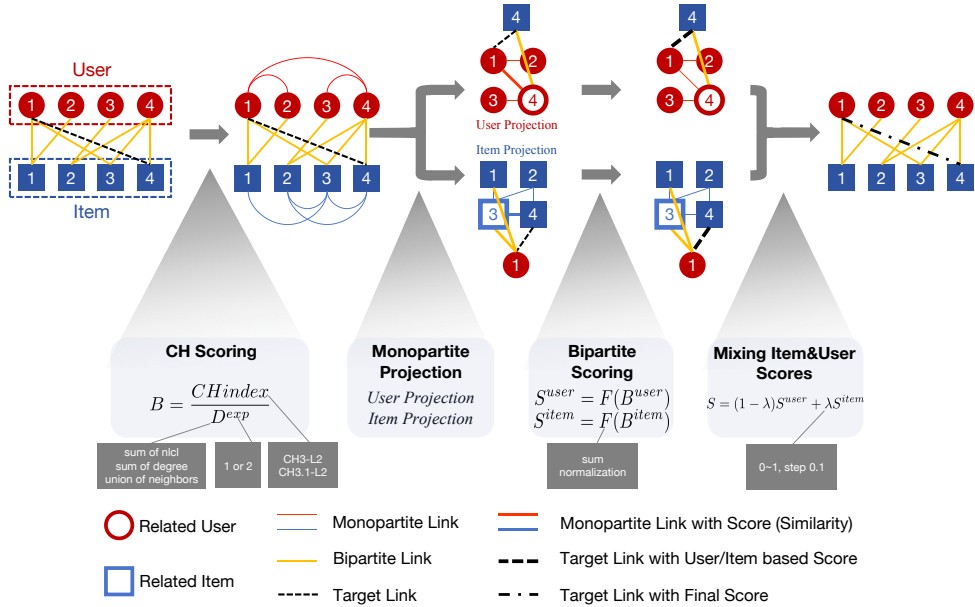

Figure 1: **Workflow of NSA.** This figure illustrates how NSA computes the prediction score for a target link through four stages: (1) CH Scoring, assigning similarity using the CH index; (2) Monopartite Projection, mapping topology and weights from the bipartite graph to two monopartite networks; (3) Bipartite Scoring, aggregating similarity into single-view recommendation scores; (4) Mixing Item and User Scores, combining single-view scores into the final prediction. *Core functions are shown in each stage block, with surrounding gray boxes indicating configurable options.

CH3.1-L2 is the version that preserves the internal-link reward when node adheres to the CH principle (low external degree), but progressively down-weights and eventually ignores this reward as node's external degree increases, signaling a violation of the CH principle. The formula of CH3.1-L2 is

$$\text{CH3.1-L2(i, j)} = \sum_{k \in L2} \frac{di_k + 1}{(1 + de_k)^{1 + \frac{de_k}{1 + de_k}}} \tag{3}$$

For clarification, $i$ and $j$ represent two nodes of the same kind in the bipartite network, $L2$ denotes the set of nodes on the path of length 2 between nodes $i$ and $j$ (specifically, in the case of a path of length 2, this can be understood as the set of common neighbors between $i$ and $j$). $de_k$ represents the number of external community links for node $k$ (i.e., the number of neighbors of node $k$ that are not in the $L2$ set and are not $i$ or $j$). $di_k$ represents the number of internal community links for node $k$ (i.e., the number of neighbors of node $k$ that are also in the $L2$ set).

**Denominator** Inspired by the weighting methods used in bipartite network projection, we introduce a scaling factor as the denominator of the CH index to reduce the weight of links connecting two nodes with many neighbors. We adopt three options as denominator with different topological meaning, including:

- **sum of degree**: the sum of the degrees of the two seed nodes

$$D_{ij} = d_i + d_j \tag{4}$$

- **union of neighbors**: the total number of neighbor nodes of the two seed nodes

$$D_{ij} = d_i + d_j - CN_{ij} \tag{5}$$

- **sum of nlcl**: the number of non local community links (nlcl) of the two seed nodes (i.e., the number of neighbors of the two seed nodes that are not in the $l2$ set)

$$D_{ij} = d_i + d_j - 2CN_{ij} \tag{6}$$

**Exponent**  To further control the impact of the scaling factor, we introduce a new exponent variable for the denominator. As the name suggests, the exponent serves as the power of the denominator base that ranges from 1 to 2.

The similarity between a pair of nodes can be represented as

$$B_{ij} = \frac{\text{CH-index(i, j)}}{D_{i,j}^{exp}} \tag{7}$$

## 3.2  MONOPARTITE PROJECTION

The CH-scoring process is based on the local community defined by paths of length two. This naturally gives rise to two projected monopartite networks, one for users and one for items, where a link exists between two nodes of the same type only if they share at least one common neighbor. We employed bipartite network projection to transform both the topology and the assigned weights into two separate monopartite networks.

## 3.3  BIPARTITE SCORING

Based on two monopartite networks respectively, we can weight all the non-existing links inside the original bipartite network based on simple aggregation. Here, we adopted two options for the aggregation:

- sum

$$S_{u\gamma}^{user} = \sum_{i=1}^{|U|} B_{ui} M_{i\gamma}, \ S_{u\gamma}^{item} = \sum_{\alpha=1}^{|\Gamma|} B_{\gamma\alpha} M_{u\alpha} \tag{8}$$

- normalization

$$S_{u\gamma}^{user} = \frac{\sum\limits_{i=1}^{|U|} B_{ui} M_{i\gamma}}{\sum\limits_{i=1}^{|U|} B_{ui}}, S_{u\gamma}^{item} = \frac{\sum\limits_{\alpha=1}^{|\Gamma|} B_{\gamma\alpha} M_{u\alpha}}{\sum\limits_{\alpha=1}^{|\Gamma|} B_{u\alpha}} \tag{9}$$

## 3.4  MIXING ITEM AND USER SCORES

Based on the item-based score and user-based score we draw from two monopartite networks, take the weighted average of item-based score and user-based score controlled by parameter $\lambda$ as the weight of item-based score, which is ranged from 0 to 1 step by 0.1. The final recommendation score can be represented as:

$$S_{u\gamma} = (1 - \lambda)S_{u\gamma}^{user} + \lambda S_{u\gamma}^{item} \tag{10}$$

## 4  EXPERIMENTS

We've conducted quite a lot of experiments to prove that our method is of superiority compared to both traditional memory-based methods and the advanced model-based methods.

### 4.1  BASELINES

The baselines adopted in our study span from traditional memory-based approaches to state-of-the-art model-based methods. Memory-based methods follow a standard pipeline and the key distinction among various memory-based approaches lies in the choice of similarity metric. Our implementation of memory-based collaborative filtering strictly follows the framework introduced in previous work (Albora et al., 2023). We selected two representative similarity measures to construct memory-based baselines: the state-of-the-art Sapling Similarity and the widely-used Jaccard Similarity, with the memory-based method built upon called SSCF and JCF respectively. We select a series of representative and state-of-the-art model-based methods as baselines, including NGCF (Wang et al., 2019), LightGCN (Mao et al., 2021a), SimpleX (Mao et al., 2021a), UltraGCN Mao et al. (2021b), LT-OCF (Choi et al., 2021), BSPM (Choi et al., 2023) and XSimGCL (Yu et al., 2023). For BSPM, to be specific, we utilized the variant BSPM-EM which offers better performance (Choi et al., 2023).

## 4.2 DATASETS

We employed 13 datasets from different filed ranging from drug-target network in biological field to typical recommendation datasets in social system (Coscia et al., 2013; Albora et al., 2023; Ruggles et al., 1995; Yildirim & Coscia, 2014; Yamanishi et al., 2008; Balassa, 1965; Hu & Bajorath, 2014; Cheng et al., 2019; McAuley et al., 2015; Pasricha & McAuley, 2018). The statistics of all the datasets are listed in Appendix B, where we reported the source, number and type of nodes and the density.

It is important to note that, for some datasets, there is no explicit distinction between users and items. For example, in drug-target networks, recommendations can be made from either the drug perspective or the target perspective, both of which are meaningful in real-world applications. Therefore, we conducted experiments from both perspectives, treating different sets of nodes as the "user" side.

## 4.3 HYPERPARAMETER LEARNING AND EVALUATION

In this section, we introduce the way we split the datasets as train and test set, the metric we used for evaluation, and the evaluation process. For clarity, the entire procedure is also illustrated in a figure provided in Appendix E.

**Metrics**    To better evaluate the performance of models, we utilized widely used metrics in recommendation system field: Recall@10, Recall@20, NDCG@10, NDCG@20.

**Train-Test Split**    We follow the widely used way to split each dataset to train set and test set. For all the datasets, the train set retains 80% links for each user randomly. The rest links would become test set which is used to evaluate the performance of models. We repeat the split several times which can be called as different *realizations* in case that the randomness of segmentation influences the evaluation of performance.

**Hyperparameter Learning**    We adopted multiple *validation samplings* to learn the most appropriate hyperparameter setting for each realization automatically. To be specific, we'll further split the train set to two parts. 10% links of each user would be randomly removed to verify the performance of different hyperparameter settings. Also, to avoid randomness, we repeat this procedure 10 times and the hyperparameter setting with highest average performance would be the one used for test. It needs attention that, when evaluated by different metrics, the best hyperparameter setting can be different. To ensure the fairness of comparison, we conducted the same hyperparameter choosing strategy on all baseline methods mentioned above strictly and carefully. The concrete hyperparameter setting under search for each baseline method are reported in Appendix D.

**Evaluation Process**    Each model would give a ranking of all the non-existing links for all the users based on the existing links in the train set, then the links with highest ranking would become the result of prediction. For each user, we would compute metrics Recall@20, Recall@10, NDCG@20 and NDCG@10. For each metric, the final performance is the average among all the users. Results reported are the average across all realizations.

## 5 RESULTS

In this section, we present a comprehensive summary, comparison, and analysis of the performance of NSA and selected baseline methods across 13 datasets. Specifically, experiments were conducted using 10 realizations under the default ViewA, and 5 realizations under the alternative ViewB. For the latter, we selected the top-performing method from each category based on the results in ViewA: NSA for memory-based methods; LT-OCF for neural network-based methods, which are considered a subset of model-based approaches; and BSPM for diffusion-based methods, which has shown competitive performance in prior work (Albora et al., 2023). Due to space limitations, additional results are provided in the Appendix.

**ViewA Results**    We present results for all methods based on individual network from ViewA in Appendix F, where NSA consistently outperforming most methods on the majority of datasets compared to a comprehensive set of baselines. To provide a comprehensive comparison, we further compute the average ranking of each method across all datasets. As shown in Fig 2 , NSA consistently ranks first or second across various metrics, highlighting its overall superiority. This consistent top-tier performance not only reflects NSA's high accuracy but also underscores its robustness and adaptability across different domains and evaluation criteria.

**ViewB Results**    NSA achieves the best average ranking across three evaluation metrics, outperforming BSPM and LT-OCF, as shown in Fig 3. The results for ViewB organized by individual networks

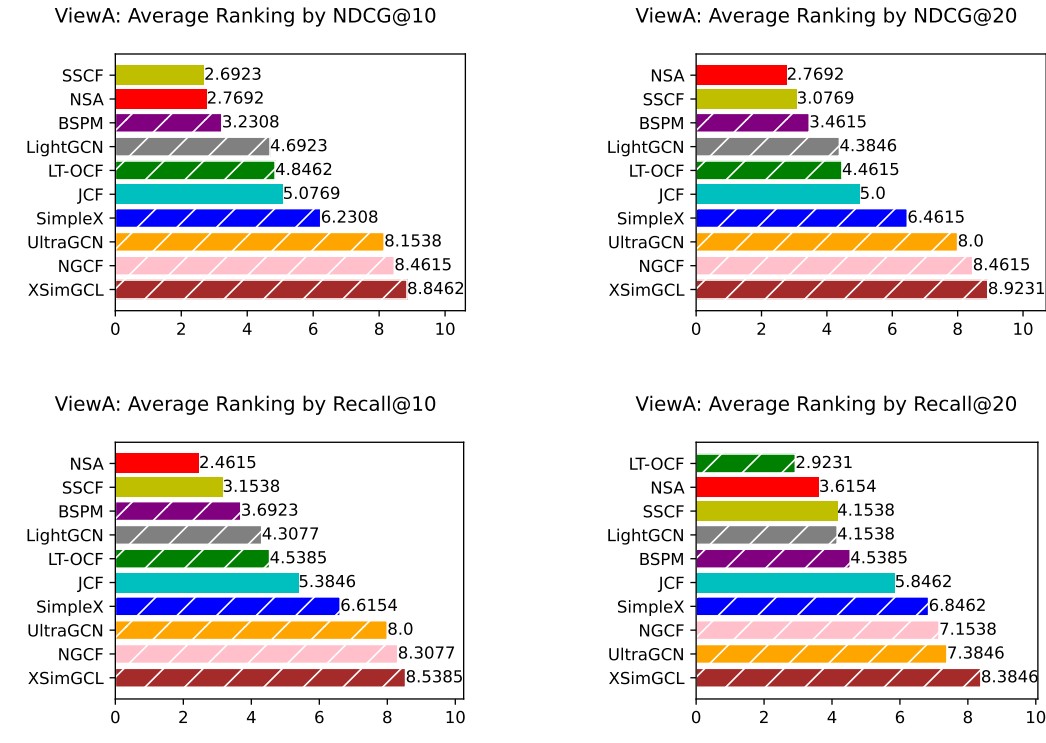

Figure 2: **ViewA: Average ranking** across 13 datasets evaluated by different metrics. To better distinguish between memory-based and model-based methods, all bars corresponding to model-based approaches are overlaid with white hatching. *ViewA means that these experiments are conducted treating nodes in set A as users.

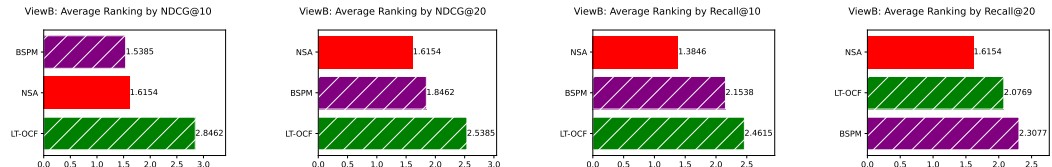

Figure 3: **ViewB: Average ranking** across 13 datasets evaluated by different metrics. Bars corresponding to model-based approaches are overlaid with white hatching. *ViewB means that these experiments are conducted treating nodes in set B as users.

are provided in Appendix G. These results further demonstrate the effectiveness and robustness of NSA.

**Robustness of NSA on Large-scale Datasets**   To evaluate the scalability of NSA, we conducted experiments on three large-scale recommendation datasets: Gowalla, Yelp2018, and Amazon-Book. Due to time constraints, we include only LT-OCF and SSCF as baselines, as these are accessible methods with average rankings comparable to or better than NSA on medium-scale datasets, as reported in Figure 2. We directly adopt the train-test splits provided in the literature, using the latest 20% of emerging links as the prediction targets. Validation is performed three times to select appropriate hyperparameter settings. For Amazon-Book, due to computational limits, we evaluate NSA using only the CH3-L2 index, referred to as NSA3. The results are reported in Table 2. Across all three datasets and various evaluation metrics, NSA demonstrates clear advantages in stability, consistently ranking first or second, while the third-place method lags significantly behind. In contrast, although LT-OCF and SSCF occasionally achieve the top rank, their performance is highly

Table 2: **Recommendation performance on large-scale datasets.** The table contains the performance of NSA compared with LT-OCF and SSCF on three classicale large-scale recommendation datasets evluated by different metrics. NSA3 refers to the variant of NSA which utilize only the CH3-L2 index.

| Dataset | Method | Recall@10 | Recall@20 | NDCG@10 | NDCG@20 |
|---|---|---|---|---|---|
| Gowalla | LT-OCF | 0.1278 | 0.1817 | **0.1370** | **0.1530** |
| | SSCF | 0.1213 | 0.1775 | 0.1209 | 0.1390 |
| | NSA | **0.1294** | **0.1864** | 0.1335 | 0.1508 |
| | Ranking of NSA | 1 | 1 | 2 | 2 |
| Yelp2018 | LT-OCF | 0.0345 | 0.0597 | 0.0388 | 0.0484 |
| | SSCF | 0.0385 | 0.0656 | 0.0442 | 0.0540 |
| | NSA | **0.0409** | **0.0684** | **0.0467** | **0.0566** |
| | Ranking of NSA | 1 | 1 | 1 | 1 |
| Amazon-book | LT-OCF | 0.0229 | 0.0400 | 0.0234 | 0.0304 |
| | SSCF | **0.0500** | **0.0773** | **0.0547** | **0.0647** |
| | NSA3 | 0.0452 | 0.0723 | 0.0491 | 0.0602 |
| | Ranking of NSA3 | 2 | 2 | 2 | 2 |

dataset-dependent. These results provide strong evidence for the robustness and effectiveness of NSA on large-scale datasets, regardless of the underlying structure or organization of the data.

**Effectiveness of a Simplified NSA Variant**   To further evaluate the flexibility and robustness of NSA, we conducted an ablation study in which the exponent parameter was fixed to a constant value of 1 without tuning in the validation stage. Interestingly, it still achieved impressive results from ViewA, ranking first on average evaluated by multiple metrics. The detailed results are provided in Appendix H. This finding demonstrates the strong performance of NSA even under a more constrained configuration.

**Training-Free Robustness of NSA**   While NSA achieves strong overall performance, we observe that it performs slightly less competitively than certain model-based methods specifically on Recall@20. This discrepancy may stem from the epoch selection strategies commonly employed by model-based approaches. In contrast, NSA is a non-training method and thus does not involve such metric-specific tuning so that it avoids potential bias introduced by overfitting and maintains consistently strong performance across other key metrics. This distinction highlights NSA's ability to preserve ranking fidelity and generalize effectively across evaluation settings, without the need for iterative optimization or metric-dependent parameter tuning.

**High Sparsity Robustness of NSA**   Especially on datasets under high sparsity level, NSA demonstrates strong performance compared to other methods. This indicates that NSA is more robust to networks with higher sparsity. Its advantage may stem from the incorporation of CH theory from network science, which enables it to extract more informative signals from the inherently sparse structures found in real-world networks.

## 6   CONCLUSION AND DISCUSSION

In this paper, we propose Network Shape Automata (NSA), a novel memory-based collaborative filtering method that leverages bipartite network topology for recommendation. Building on recent progress in memory-based methods, NSA further explores the potential of this class of approaches, emphasizing simplicity, interpretability, and strong performance. NSA introduces the Cannistraci-Hebb (CH) theory from network science as the foundation for its similarity measure. This theory, inspired by the evolution of brain neural networks, enables NSA to utilize local community structures based on topological features of real-world networks, without requiring any training. We evaluate NSA on 13 real-world bipartite datasets across multiple domains and compare it against both memory-based and model-based collaborative filtering methods. We conducted experiments on networks with up to 9,865 nodes and 172,206 edges. For ViewA alone, we performed thorough hyperparameter learning and evaluation on 13 networks using 9 methods, each with 3 hyperparameters (averaging 9 settings), across 10 realizations with 10 validation samplings, resulting in a total of 105,300

model assessments. Experimental results show that NSA consistently outperforms strong baselines across multiple evaluation metrics. It also demonstrates notable robustness under high sparsity and maintains stable and superior performance on large-scale datasets, while preserving the desirable traits of memory-based approaches. Overall, NSA highlights the overlooked potential of memory-based collaborative filtering in modern recommendation systems and validates the effectiveness of the Cannistraci-Hebb theory in modeling network evolution for link prediction and recommendation tasks.

REPRODUCIBILITY STATEMENT

The code for this work is provided in the supplementary material. Detailed hyperparameter settings for each method are presented in Appendix D to facilitate reproducibility.

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

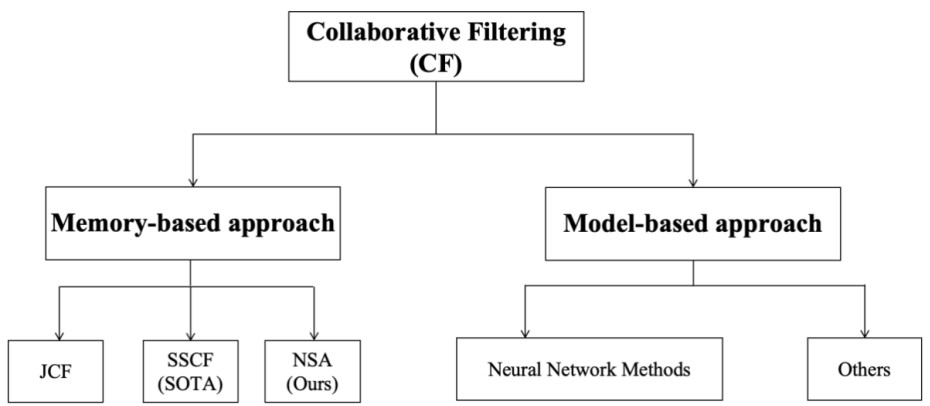

Figure 4: The classification of Collaborative Filtering.

Table 3: Statistics of Datasets

| Index | Name | Field | TypeA | #NodeA | TypeB | #NodeB | #Link | Density |
|---|---|---|---|---|---|---|---|---|
| **D1** | aidorganizations_issues Coscia et al. (2013) | Social | orgnization | 151 | issue | 34 | 1889 | 36.79% |
| **D2** | export Balassa (1965) | Social | country | 169 | item | 4957 | 120377 | 14.37% |
| **D3** | industries_educationfields_IPUMS Ruggles et al. (1995) | Social | industry | 267 | education | 513 | 18088 | 13.21% |
| **D4** | congressmen_topics_US Yildirim & Coscia (2014) | Social | congressmen | 525 | topic | 970 | 56215 | 11.04% |
| **D5** | users_movies_movielens100k | Social | user | 943 | movie | 1574 | 82520 | 5.56% |
| **D6** | drug_target_ionchannel_2009 Yamanishi et al. (2008) | Biological | drug | 210 | target | 204 | 1476 | 3.45% |
| **D7** | drug_target_GPCR_2009 Yamanishi et al. (2008) | Biological | drug | 223 | target | 95 | 635 | 3.00% |
| **D8** | occupations_tasks_ONET Yildirim & Coscia (2014) | Social | occupation | 428 | task | 1691 | 16936 | 2.34% |
| **D9** | tfs_genes_regulation_ecoli | Biological | protein | 212 | gene | 1856 | 4496 | 1.14% |
| **D10** | amazon-product McAuley et al. (2015); Pasricha & McAuley (2018) | Social | user | 6121 | item | 2744 | 172206 | 1.03% |
| **D11** | drug_target_enzyme_2009 Yamanishi et al. (2008) | Biological | drug | 445 | target | 664 | 2926 | 0.99% |
| **D12** | drug_target_HQ_2014 Hu & Bajorath (2014) | Biological | drug | 518 | target | 358 | 1666 | 0.90% |
| **D13** | drug_target_moesm4_esm Cheng et al. (2019) | Biological | drug | 4428 | target | 2256 | 15051 | 0.15% |

# A    CLASSIFICATION OF COLLABORATIVE FILTERING

In Fig 4, we illustrate that collaborative filtering can be further divided into memory-based and model-based method. NSA can be classified as a memory-based approach.

# B    STATISTICS OF DATASETS

In this section, we present the detailed statistics of all the datasets we used for test. In Table 3, we summarized the source, number of nodes, type of nodes and density of each dataset. For clarity, we give each dataset an index in descending order considering network density.

# C    EXPERIMENTAL ENVIRONMENT

The NSA experiments are conducted in a CPU-based computing environment equipped with an AMD processor featuring 64 cores, using MATLAB and C++. The number of CPU cores employed during execution is configurable, allowing flexible adaptation to the available computational resources.

## D    HYPERPARAMETER SETTING

For all the baseline methods we're using, we listed all the hyperparameters we used for experiments in Table 4 (the rows with yellow background refer to the tuned parameters). For memory-based methods, there's limited range for hyperparameters to tune. For model-based methods, we chose the appropriate range of hyperprameter based on what mentioned in literature and preliminary experiments for each dataset.

Table 4: Hyperparameters For Different Methods

| Classification | Algorithm | Parameter | Tuning value |
|---|---|---|---|
| Memory-based | NSA | CH index | CH3-L2, CH3.1-L2 |
| | | denominator | sum of degree, sum of nlcl, union of neighbours |
| | | exponent | 1, 2 |
| | | bipartite scoring | sum, normalization |
| | | mixing parameter | 0-1, interval 0.1 |
| | SSCF | mixing parameter | 0-1, interval 0.1 |
| | JCF | mixing parameter | 0-1, interval 0.1 |
| Model-based | NGCF | lr | 1e-3, 1e-4, 1e-5 |
| | | reg | 1e-4, 1e-5, 1e-6 |
| | | embed_size | 64 |
| | | layer size | [64, 64, 64] |
| | | batch size | 1024 |
| | | node dropout | 0.1 |
| | | mess dropout | [0.1, 0.1, 0.1] |
| | LightGCN | lr | 1e-2, 1e-3, 1e-4 |
| | | decay | 1e-3, 1e-4, 1e-5 |
| | | recdim | 64 |
| | | dropout | 0 |
| | | layer | 3 |
| | | bpr_batch | 2048 |
| | SimpleX | lr | 1e-3, 1e-4, 1e-5 |
| | | gamma | 0.8, 0.5 |
| | | negative weight | 250, 10 |
| | | embedding_dim | 64 |
| | | num neg | 1000 |
| | | margin | 0.9 |
| | | net_dropout | 0.1 |
| | | batch size | 1024 |
| | UltraGCN | lr | 1e-2, 1e-1 |
| | | gamma | 1e-3, 1e-4, 1e-5 |
| | | lambda | 5e-4, 1e-5 |
| | | batch size | 512 |
| | | negative weight | 300 |
| | | embedding dim | 64 |
| | LT-OCF | lr | 1e-2, 1e-3, 1e-4 |
| | | k | 4, 2 |
| | | decay | 1e-4 |
| | | lrt | 1e-5 |
| | BSPM | lr | 1e-3, 1e-2 |
| | | idl_betas | 0.2, 0.3 |
| | | factor_dims | 12, 50 |
| | | decay | 1e-4 |
| | | dropout | 0 |
| | | layer | 3 |
| | XSimGCL | n_layers | 1, 2, 3 |
| | | l* | 1, 3 |
| | | tau | 0.15, 0.1, 0.05 |
| | | lr | 1e-3 |
| | | reg_lambda | 1e-4 |
| | | lambda | 0.05 |
| | | epsilon | 0.2 |

# E    HYPERPARAMETER LEARNING AND EVALUATION PROCESS

To better illustrates the evaluation process, we present the whole procedure by a figure. Note that, for time reason, ViewA results are the average among 10 realizations, while ViewB results are based on 5 realizations. For each realization, we conducted 10 validation samplings to find the best hyperparameter setting. Also, for different metrics, the best hyperparameter settings can be different.

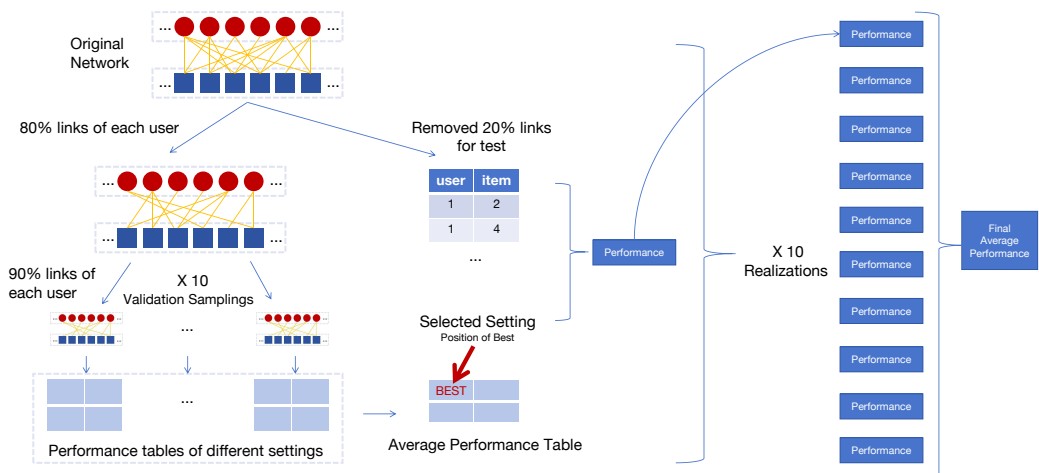

Figure 5: **Hyperparameter Learning and Evaluation procedure.** We conducted thorough hyperparameter learning and evaluation for each method according to this to get the final performance: (1) split the original dataset for different realizations; (2) for each realization, conduct 10 validation samplings to determine the best setting and then utilized it for evaluation; (3) report the final average performance across all realizations.

# F VIEWA RESULTS ON INDIVIDUAL NETWORK

For page limit, results from ViewA evaluated by different metics on each network are reported here. Since the scale of some datasets can be small, it is of significance to evaluate the performance based on both top 20 and top 10 performance. Here we can find that NSA is competitive across different metrics.

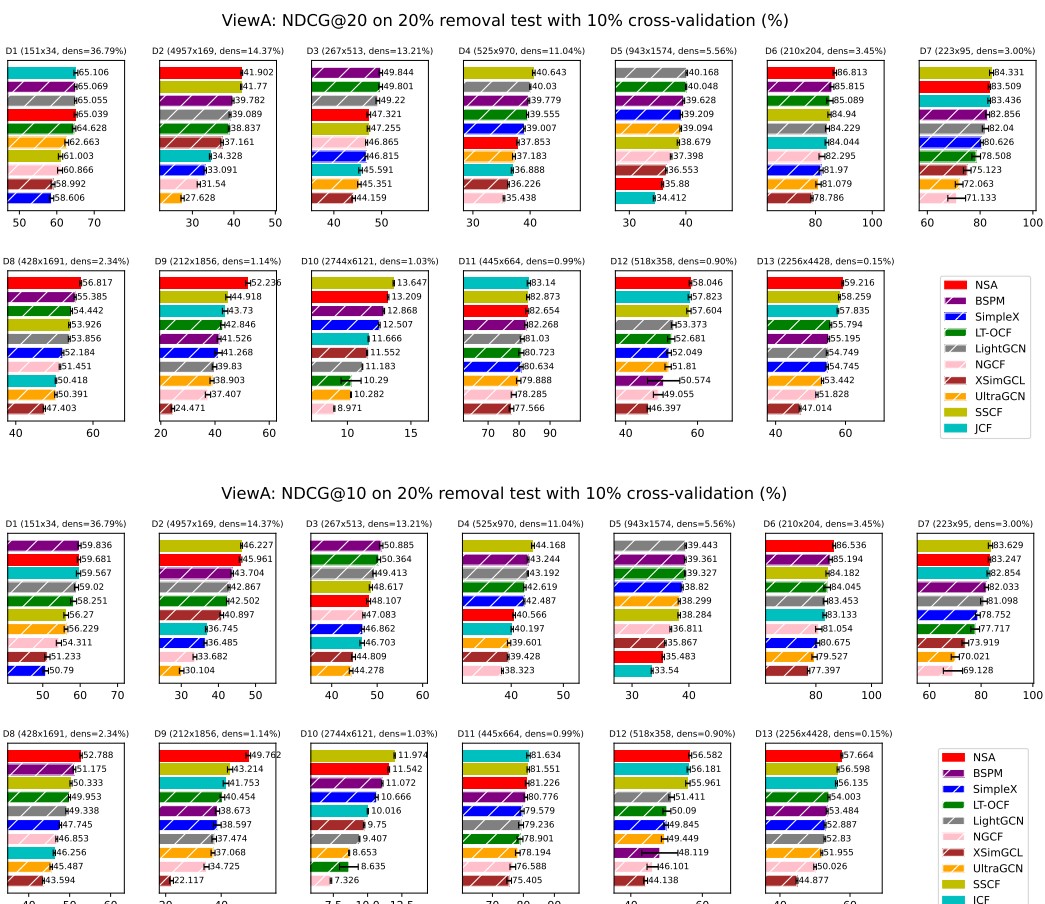

Figure 6: **ViewA: Performance evaluated by NDCG on individual dataset.** Bars corresponding to model-based approaches are overlaid with white hatching. *ViewA means that these experiments are conducted treating nodes in set A as users. Error bars represent sample standard deviation (with degrees of freedom = 1).

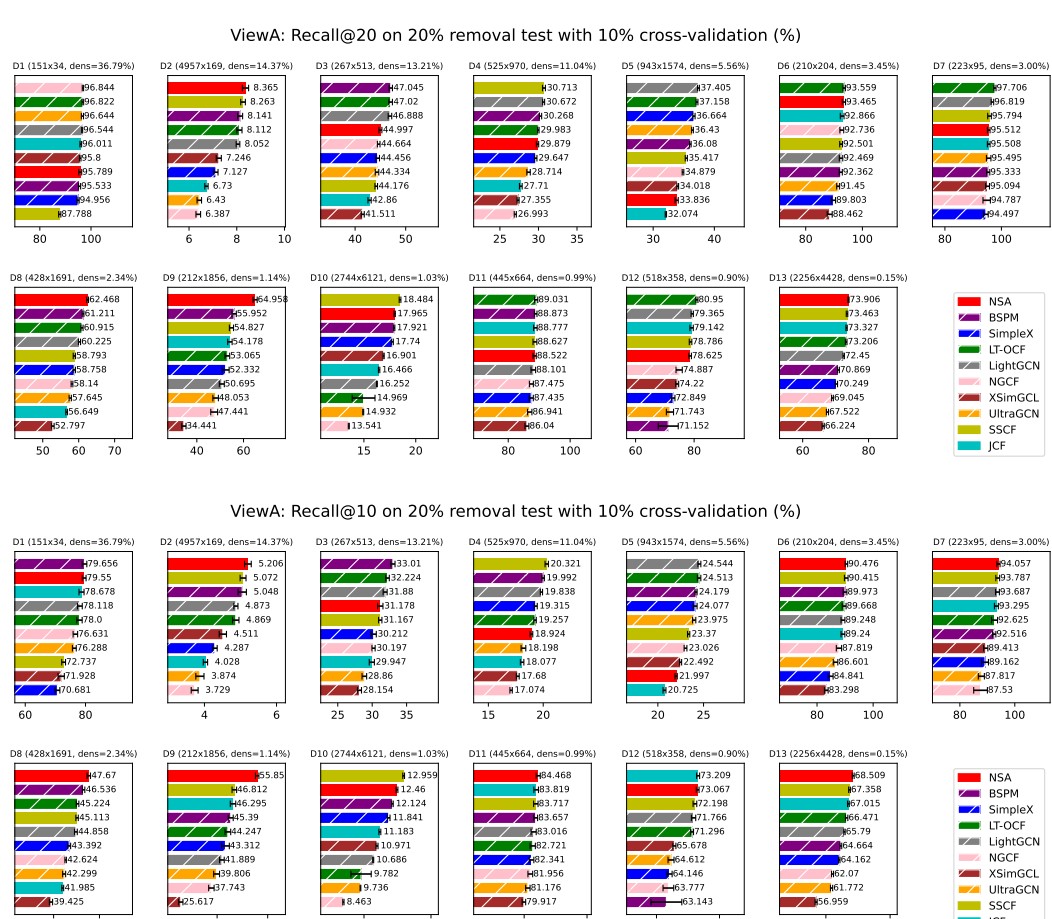

Figure 7: **ViewA: Performance evaluated by Recall on individual dataset.** Bars corresponding to model-based approaches are overlaid with white hatching. *ViewA means that these experiments are conducted treating nodes in set A as users.

# G  VIEWB RESULTS ON INDIVIDUAL NETWORK

In this section, we present the results from ViewB. For time reason, only BSPM and LT-OCF which are the two model-based methods shows the most potential from ViewA. With 5 tests repeated, NSA remains competitive on different metrics.

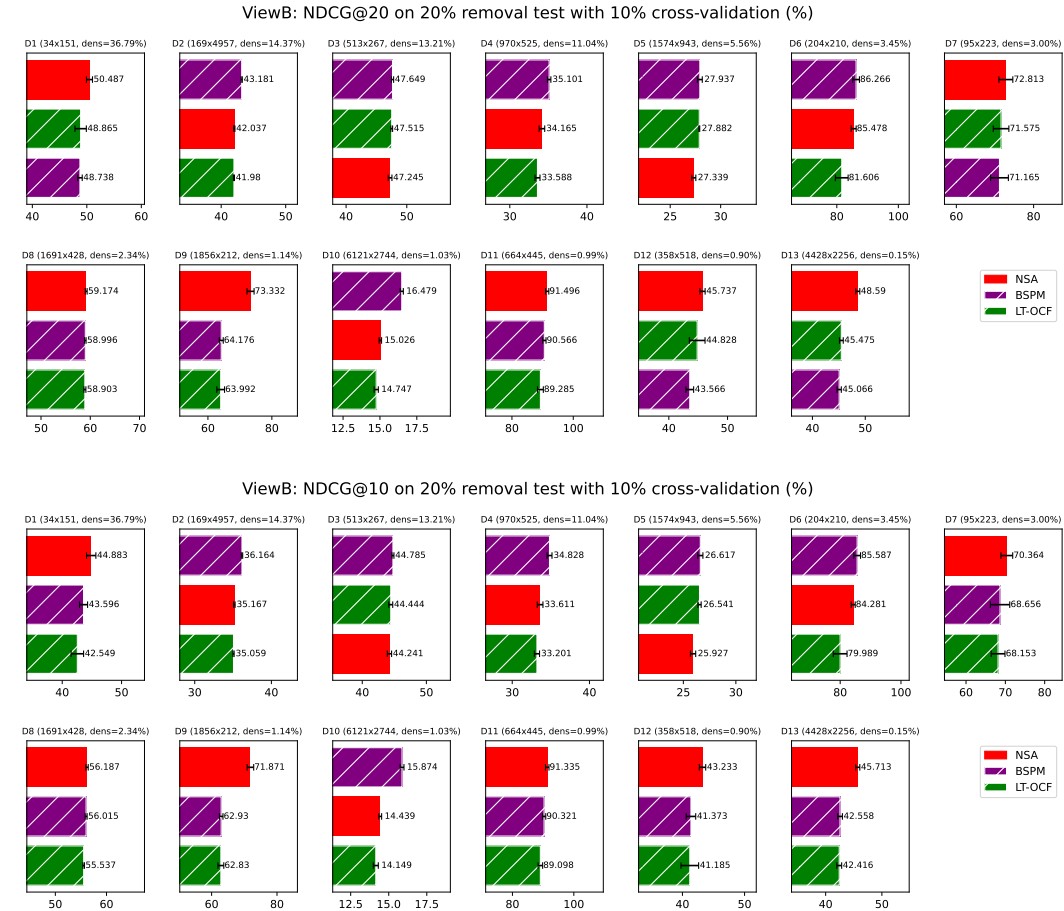

Figure 8: **ViewB: Performance evaluated by NDCG on individual dataset.** Bars corresponding to model-based approaches are overlaid with white hatching. *ViewB means that these experiments are conducted treating nodes in set B as users. Error bars represent sample standard deviation (with degrees of freedom = 1).

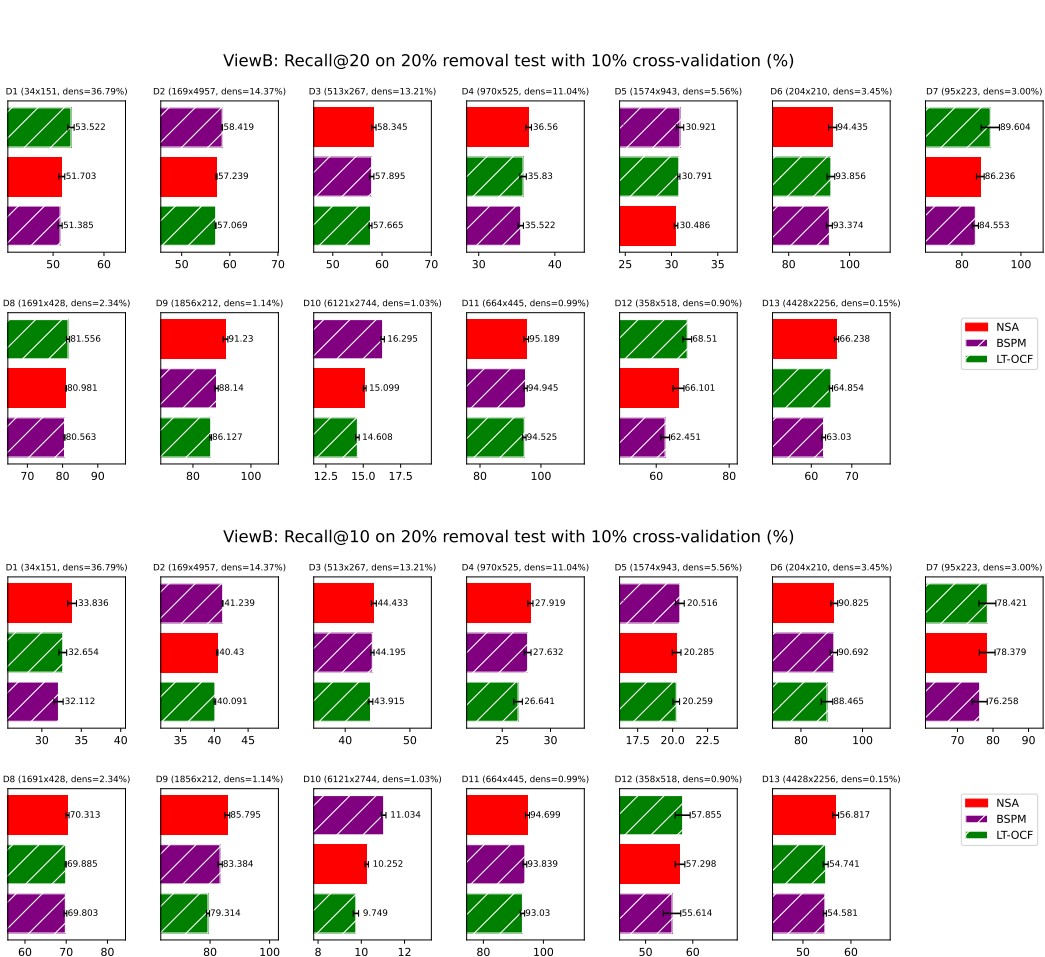

Figure 9: **ViewB: Performance evaluated by Recall on individual dataset.** Bars corresponding to model-based approaches are overlaid with white hatching. *ViewB means that these experiments are conducted treating nodes in set B as users. Error bars represent sample standard deviation (with degrees of freedom = 1).

## H  NSA WITH FIXED EXPONENT 1 RESULTS FROM VIEWA

In this section, we reported the results of simplified version NSA, with its configurable exponent being fixed to 1. Surprisingly we found that it performs quite well, with its average ranking consistently being the first across all the metrics we test.

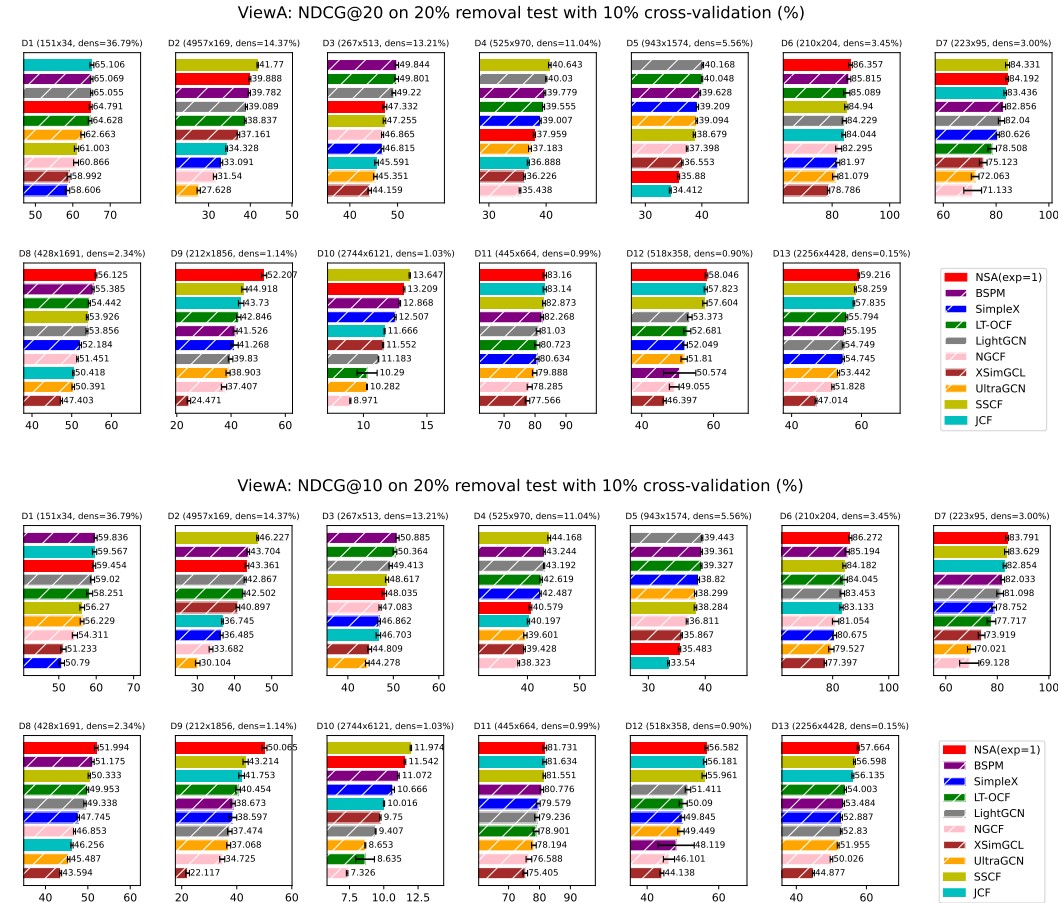

Figure 10: **ViewA: NSA(exp=1) Performance evaluated by NDCG on individual dataset.** Bars corresponding to model-based approaches are overlaid with white hatching. *ViewA means that these experiments are conducted treating nodes in set A as users. Error bars represent sample standard deviation (with degrees of freedom = 1).

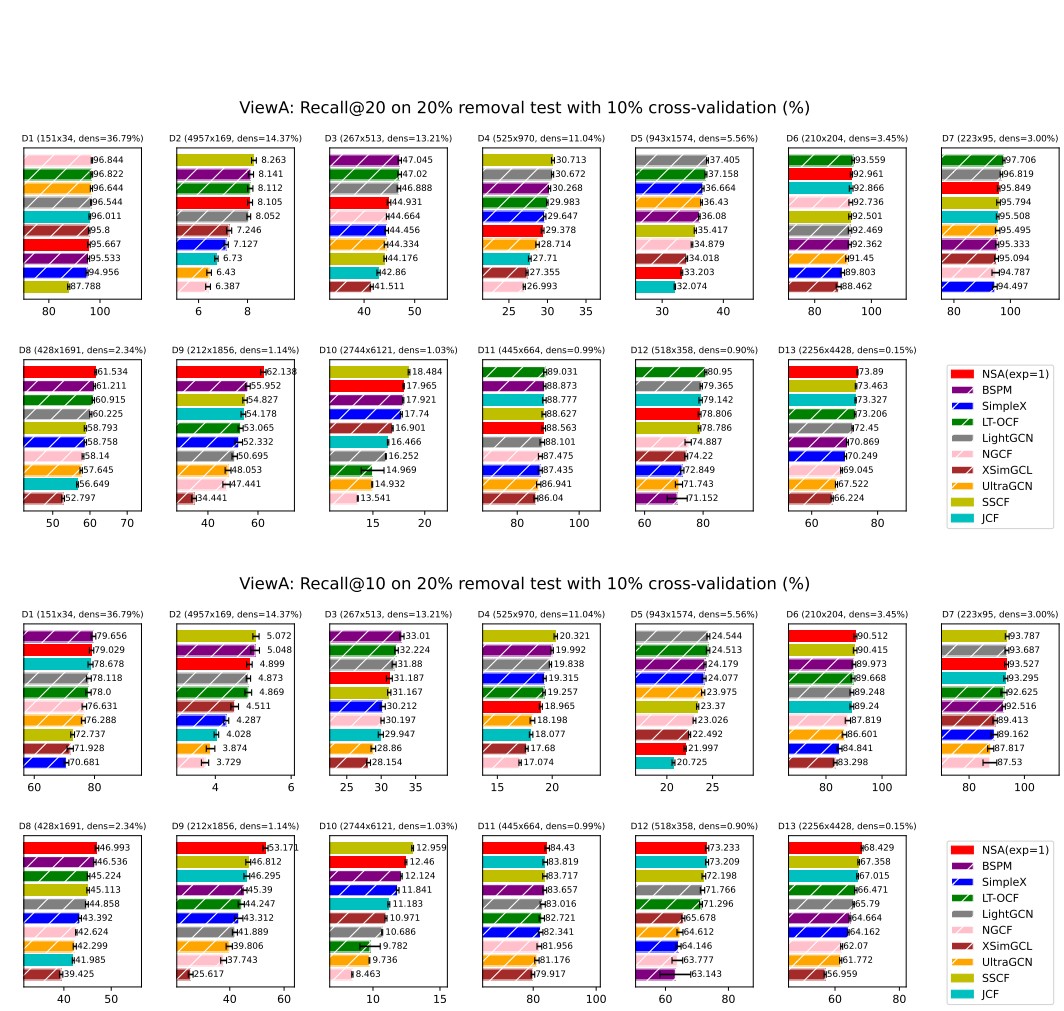

Figure 11: **ViewA: NSA(exp=1) Performance evaluated by Recall on individual dataset.** Bars corresponding to model-based approaches are overlaid with white hatching. *ViewA means that these experiments are conducted treating nodes in set A as users. Error bars represent sample standard deviation (with degrees of freedom = 1).

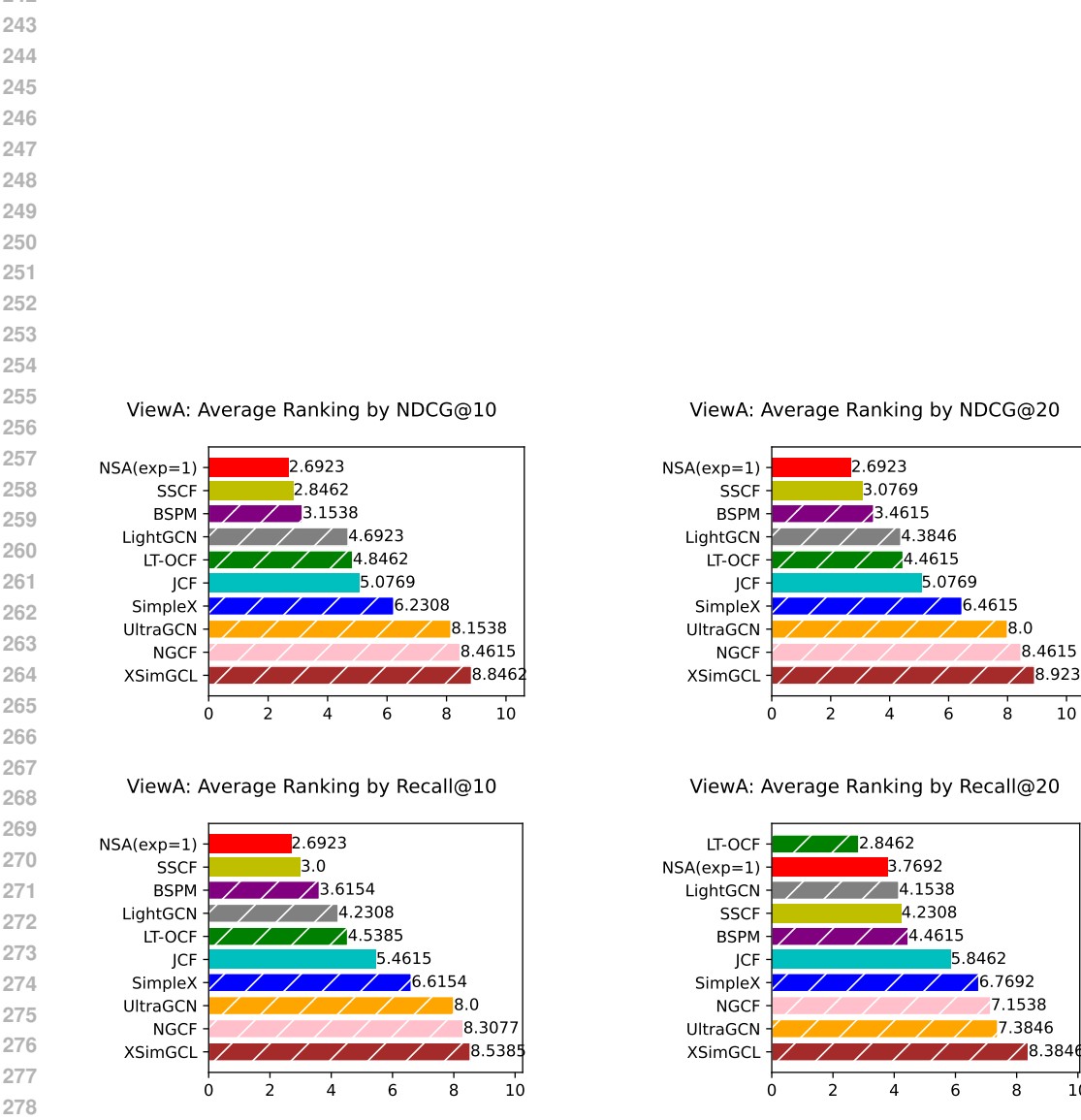

Figure 12: **ViewA: NSA(exp=1) Average ranking** across 13 datasets evaluated by different metrics. Bars corresponding to model-based approaches are overlaid with white hatching. *ViewB means that these experiments are conducted treating nodes in set B as users.

## I    BROADER IMPACT AND FUTURE WORK

**Broader Impact**    NSA is a link prediction model applicable to recommendation systems and network modeling tasks. Its simplicity makes it both interpretable and easy to implement and integrate into existing infrastructures. Potential real-world applications include personalized content delivery and modeling social connections (e.g., friend suggestions on social platforms). However, like other link prediction models, NSA may unintentionally amplify existing biases or propagate misinformation, particularly when deployed without proper safeguards. To mitigate such risks, practitioners should regularly audit model outputs, monitor their downstream impact in live environments, and incorporate human feedback mechanisms to ensure responsible use.

**Future Work**    NSA is built upon the principles of memory-based methods, which, while effective and offering higher interpretability, can be sensitive to network scale, as they often require access to the entire dataset to aggregate interaction information. In contrast, model-based methods offer better scalability through iterative processing and compact representations. Future work could focus on combining NSA with model-based techniques to enhance scalability, exploring sampling strategies to reduce memory consumption, and developing online or incremental variants of NSA that are suitable for streaming or dynamically evolving networks. Furthermore, investigating NSA's robustness and fairness under adversarial or biased conditions would further strengthen its practical applicability.

# J    TIME COMPLEXITY OF NSA

In this section, we'll explain the time complexity of our method NSA. We'll start with the basic definition and explain the time complexity step by step.

## J.1    BASIC DEFINITION

- $U$: number of users
- $I$: number of items

## J.2    CH SCORING AND MONOPARTITE PROJECTION

**CH index**    The time complexity of CH index on path of length 2 computation is determined by the cost of computing iLCL and eLCL statistics for the intermediate nodes along those paths. Here we'll discuss the time complexity in a general case, where $n$ and $m$ denote the number of nodes and edges in a network, respectively. $\bar{d} = 2m/n$ is the average degree.

- **Path count.** Each length-2 path $u \to z \to v$ is defined by an intermediate node $z$ connected to both $u$ and $v$. The total number of such paths is given by:

$$\#\text{L2\_path} = \sum_{z=1}^{n} \binom{d_z}{2} = \sum_{z=1}^{n} \frac{d_z(d_z - 1)}{2} = \mathcal{O}\left(\sum_{z=1}^{n} d_z^2\right)$$

  where $d_z$ is the degree of node $z$. This represents the number of unique unordered two-hop paths in the network.

- **Computation per path.** For each length-2 path, CHA computes a score based on the iLCL and eLCL of the intermediate node $z$. This requires checking the neighbors of $z$ against the local community associated with the pair $(u, v)$, which takes $\mathcal{O}(d_z)$ time per path.

- **Overall time complexity.** Multiplying the path count and per-path cost gives the total time complexity:

$$\mathcal{O}\left(\sum_{z=1}^{n} d_z^2 \cdot d_z\right) = \mathcal{O}\left(\sum_{z=1}^{n} d_z^3\right)$$

We now analyze this quantity under three typical network regimes:

- *Sparse, degree-homogeneous:* If the graph is Sparse (i.e. $\bar{d} = 2m/n = \mathcal{O}(1)$) with relatively uniform degrees (i.e., $d_z = \mathcal{O}(1)$ for all $z$), then:

$$\mathcal{O}\left(\sum_{z=1}^{n} d_z^3\right) = \mathcal{O}(n)$$

  So the overall time complexity of $\mathcal{O}(n)$.

- *Sparse, degree-heterogeneous:* If the graph is sparse (i.e., $\bar{d} = \mathcal{O}(1)$), but has a skewed degree distribution (e.g., power law), we can no longer assume $d_z = \mathcal{O}(1)$ for all nodes. To handle this case, we apply a relaxation via Hölder's inequality to upper-bound the root-mean-cube degree $\left(\frac{1}{n}\sum_z d_z^3\right)^{1/3}$ in terms of the average degree:

$$\left(\frac{1}{n}\sum_{z=1}^{n} d_z^3\right)^{1/3} \leq n^{2/3} \cdot \left(\frac{1}{n}\sum_{z=1}^{n} d_z\right) = n^{2/3} \cdot \bar{d} = \mathcal{O}(n^{2/3})$$

  This relaxation allows us to express the cubic-degree term in the overall complexity as:

$$\mathcal{O}\left(\sum_{z=1}^{n} d_z^3\right) = \mathcal{O}\left(n \cdot \left(\frac{1}{n}\sum_{z=1}^{n} d_z^3\right)\right) = \mathcal{O}\left(n \cdot \left(n^{2/3}\right)^3\right) = \mathcal{O}(n^3)$$

  Thus, the overall time complexity in this case is $\mathcal{O}(n^3)$.

- *Dense graphs:* In the worst-case scenario of dense graphs, where $d_z = \mathcal{O}(n)$ for all nodes, we obtain:

$$\sum_{z=1}^{n} d_z^3 = \mathcal{O}(n^4)$$

leading to an overall time complexity of $\mathcal{O}(n^4)$.

We compute CH index on the whole bipartite network, which means that in our case, $n = U + L$.

**Denominator** The computation of denominator is related to common neighbors ($CN_{i,j}$) between two nodes of same kind. The computation of common neighbors between two nodes of same kind is implemented by the dot product of the adjacent matrix and its transpose. This procedure is offline and the results can be reused always. For user based, it's of time complexity $U^2 I$, while for item based it's of time complexity $UI^2$. Since we want the similarity score on two projected monopartite networks, we only need to consider $U^2 + I^2$ computations of denominator. The final time complexity of denominator computation can be $\mathcal{O}(U^2 I + UI^2 + U^2 + I^2) = \mathcal{O}(U^2 I + UI^2)$.

## J.3 BIPARTITE SCORING

In this step, we aggregate the similarity scores on two monopartite networks separately to the link prediction scores. For instance, when we compute the user-based link prediction score, we utilized the user similarity matrix of size $U \times U$ and adjacent matrix of size $U \times I$ utilizing sum or normalization method. For each user-item pair, we compute the score using all the user's similarity corresponding to our target user so that the complexity can be $U$ exactly. Hence, the user based link prediction complexity should be multiplied with all pairs of user-item pair and result in time complexity of $\mathcal{O}(U^2 I)$. Correspondingly the item based link prediction score time complexity can be of $\mathcal{O}(UI^2)$. The total time complexity in this step can be $\mathcal{O}(U^2 I + UI^2)$.

## J.4 MIX ITEM AND USER SCORES

For each user-item pair, we aggregate user and item score, so that the time complexity is $\mathcal{O}(UI)$

## J.5 SUMMARY

Corresponding the different network regime mentioned in section J.2, we summarize here the overall time complexity of NSA.

- *Sparse, degree-homogeneous:* The dominant component of the time complexity is the collaborative filtering mechanism, result in overall complexity of $\mathcal{O}(U^2 I + UI^2)$.
- *Sparse, degree-heterogeneous:* The dominant component of the time complexity is CH score computation, result in overall complexity of $\mathcal{O}((U + I)^3)$.
- *Dense graphs:* The dominant component of the time complexity is CH score computation, result in overall complexity of $\mathcal{O}((U + I)^4)$ which is rare for recommendation system tasks.

Table 5: **Summary of Running Time.** All reported times are averaged over three runs. Experiments for memory-based methods were conducted on an AMD Ryzen Threadripper PRO 3995WX CPU with 64 physical cores, while other methods were conducted on an NVIDIA RTX A6000 GPU. All time values are expressed in seconds (s).

| Dataset | NSA | SSCF | JCF | NGCF | LightGCN | UltraGCN | SimpleX | LT-OCF | BSPM |
|---|---|---|---|---|---|---|---|---|---|
| aidorganizations_issues | 0.06± 0.00 | 0.06± 0.00 | 0.06± 0.00 | 28.90± 1.52 | 11.30± 0.07 | 35.80± 0.74 | 22.39± 0.42 | 13.41± 0.09 | 17.31± 0.27 |
| export | 5.40± 0.01 | 1.55± 0.00 | 1.20± 0.00 | 1129.81± 8.55 | 435.42± 3.44 | 277.43± 2.16 | 267.52± 0.63 | 617.89± 31.94 | 22.75± 0.03 |
| industries_eductionfields_IPUMS | 0.35± 0.00 | 0.34± 0.00 | 0.40± 0.00 | 150.53± 2.49 | 67.50± 0.54 | 75.34± 0.37 | 34.17± 0.94 | 96.63± 0.85 | 17.99± 0.13 |
| congressmen_topics_US | 1.21± 0.01 | 1.24± 0.00 | 1.41± 0.02 | 340.44± 1.96 | 209.35± 2.30 | 157.58± 1.26 | 113.68± 3.61 | 269.45± 4.39 | 18.97± 0.20 |
| users_movies_movielens100k | 2.90± 0.00 | 3.15± 0.01 | 3.65± 0.01 | 477.04± 5.27 | 288.40± 0.68 | 205.45± 2.18 | 132.38± 2.65 | 402.87± 2.50 | 19.33± 0.07 |
| drug_target_ionchannel_2009 | 0.13± 0.00 | 0.13± 0.00 | 0.12± 0.00 | 70.05± 2.24 | 9.85± 0.37 | 35.58± 0.70 | 12.84± 0.41 | 11.10± 0.27 | 17.71± 0.04 |
| drug_target_GPCR_2009 | 0.09± 0.00 | 0.09± 0.00 | 0.09± 0.00 | 39.53± 1.36 | 6.98± 0.06 | 34.92± 1.14 | 13.85± 0.95 | 8.58± 0.13 | 17.87± 0.13 |
| occupations_tasks_ONET | 1.32± 0.00 | 1.51± 0.00 | 1.76± 0.01 | 141.58± 0.41 | 61.06± 0.37 | 76.30± 0.47 | 63.22± 2.62 | 83.36± 0.36 | 17.91± 0.12 |
| tfs_genes_regulation_ecoli | 0.65± 0.00 | 1.00± 0.00 | 0.57± 0.01 | 82.75± 0.23 | 19.40± 0.13 | 41.33± 0.54 | 24.49± 0.93 | 23.85± 0.11 | 18.04± 0.05 |
| amazon-product | 26.81± 0.12 | 25.67± 0.05 | 25.25± 0.14 | 924.45± 6.48 | 600.66± 1.88 | 385.63± 1.44 | 394.01± 2.27 | 836.37± 4.80 | 22.19± 0.05 |
| drug_target_enzyme_2009 | 0.37± 0.00 | 0.54± 0.00 | 0.30± 0.00 | 64.72± 0.05 | 14.46± 0.20 | 39.43± 1.12 | 12.96± 0.24 | 19.76± 0.14 | 17.70± 0.11 |
| drug_target_HQ_2014 | 0.32± 0.00 | 0.43± 0.00 | 0.30± 0.00 | 67.52± 0.36 | 10.33± 0.12 | 35.92± 1.32 | 18.43± 0.59 | 12.29± 0.27 | 17.83± 0.09 |
| drug_target_moesm4_esm | 12.30± 0.10 | 16.36± 0.04 | 11.66± 0.01 | 135.00± 2.90 | 60.33± 0.11 | 74.57± 0.37 | 58.64± 1.48 | 85.20± 0.63 | 18.85± 0.07 |

Table 6: **Running time of NSA with different number of configurations tested.** The NSA (avg. over settings) column reports the running time of NSA when multiple configurations tested. The value reported is based on second (s), 3 times average with standard error.

| Dataset | NSA | NSA (avg. over settings) |
|---|---|---|
| D1 | 0.06±0.00 | 0.04±0.00 |
| D2 | 5.40±0.01 | 0.89±0.00 |
| D3 | 0.35±0.00 | 0.26±0.00 |
| D4 | 1.21±0.01 | 0.89±0.00 |
| D5 | 2.90±0.00 | 2.35±0.01 |
| D6 | 0.13±0.00 | 0.09±0.00 |
| D7 | 0.09±0.00 | 0.07±0.00 |
| D8 | 1.32±0.00 | 1.15±0.00 |
| D9 | 0.65±0.00 | 0.48±0.00 |
| D10 | 26.81±0.12 | 21.52±0.05 |
| D11 | 0.37±0.00 | 0.27±0.00 |
| D12 | 0.32±0.00 | 0.24±0.00 |
| D13 | 12.30±0.10 | 9.95±0.02 |

## K    EXPERIMENTAL TIME

In this section, we listed the running time of each method on different datasets with one hyperparameter setting in Table 5.

Moreover, NSA, as a memory-based method, benefits from reusing computational components efficiently during validation. To be specific, when searching hyperparameter settings, instead of training from scratch several times as model-based methods, NSA could simply reuse the components when changing hyperparameters like the mixing parameter. Statistically, we report the comparison in Table 6 between time of NSA with 1 configuration tested and average time of NSA with 10 configurations tested, which is often the case during validation. It shows that the running time averaged over settings of NSA is improved as expected for the component reusage which would benefit a lot the validation process.

Table 7: **Peak memory usage of different methods.** The reported values in the table are based on MB, measured by "/usr/bin/time -v".

|      | NSA  | SSCF | JCF  | BSPM | LightGCN | LT-OCF | NGCF | SimpleX | UltraGCN |
|------|------|------|------|------|----------|--------|------|---------|----------|
| D1   | 639  | 585  | 594  | 6352 | 5397     | 5394   | 1201 | 4958    | 2705     |
| D2   | 1955 | 684  | 674  | 6330 | 5388     | 5421   | 1726 | 10099   | 2707     |
| D3   | 717  | 586  | 582  | 6352 | 5404     | 5399   | 1236 | 5567    | 2704     |
| D4   | 828  | 591  | 592  | 6350 | 5391     | 5424   | 1281 | 7020    | 2713     |
| D5   | 1098 | 593  | 585  | 6344 | 5381     | 5408   | 1317 | 8058    | 2710     |
| D6   | 676  | 587  | 585  | 6334 | 5385     | 5403   | 1232 | 4943    | 2707     |
| D7   | 656  | 587  | 595  | 6351 | 5405     | 5400   | 1201 | 4919    | 2707     |
| D8   | 917  | 593  | 590  | 6316 | 5381     | 5397   | 1245 | 5545    | 2701     |
| D9   | 863  | 594  | 585  | 6335 | 5387     | 5411   | 1252 | 5080    | 2705     |
| D10  | 6804 | 3110 | 3112 | 6450 | 5413     | 5430   | 1737 | 11587   | 2764     |
| D11  | 754  | 586  | 586  | 6365 | 5393     | 5398   | 1230 | 5010    | 2702     |
| D12  | 724  | 586  | 594  | 6339 | 5395     | 5398   | 1236 | 4955    | 2717     |
| D13  | 3444 | 1970 | 1964 | 6343 | 5393     | 5418   | 1257 | 5502    | 2718     |

## L    TIME COMPLEXITY OF BASELINES

### L.1    DEFINITION

For clarification, all the mathematical symbol mentioned below are defined here.

- U: number of users
- I: number of items
- E: number of edges in the network
- L: number of layers for neural-network based methods
- D: dimension of embedding in model-based methods
- N: number of negative samples
- K: number of sampling similar neighbors
- T: number of epochs for neural-network based methods

### L.2    TIME COMPLEXITY

We list below the time complexities of the baseline methods, based on their respective descriptions in the original papers.

- NGCF: $\mathcal{O}\Big(T \times L \times E \times D^2\Big)$
- LightGCN: Not declared
- UltraGCN: $\mathcal{O}\Big(T \times E \times (1 + K + N) \times D^2\Big)$
- SimpleX: Not declared
- LT-OCF: Not declared
- BSPM: Not declared
- SSCF: $\mathcal{O}(U^2 I + U I^2)$
- JCF: $\mathcal{O}(U^2 I + U I^2)$

## M    MEMORY USAGE OF BASELINES

In this section, we report the memory usage of different methods in Table 7. Results show NSA consumes less memory than many model-based methods on 13 middle-scale datasets.

Table 8: **Comparison between CH theory based NSA and CN based NSA.** The table consists of performance evaluated by different metrics on different datasets of NSA based on CH theory or commen neighbors.

| | Recall@10 | | Recall@20 | | NDCG@10 | | NDCG@20 | |
|---|---|---|---|---|---|---|---|---|
| | NSA | NSA(CN) | NSA | NSA(CN) | NSA | NSA(CN) | NSA | NSA(CN) |
| aidorganizations_issues | **0.7955 ± 0.0201** | 0.7619 ± 0.0325 | **0.9579 ± 0.0183** | 0.9497 ± 0.0175 | **0.5968 ± 0.0139** | 0.5688 ± 0.0232 | **0.6504 ± 0.0129** | 0.6272 ± 0.0182 |
| congressmen_topics_US | **0.1892 ± 0.0039** | 0.0434 ± 0.0134 | **0.2988 ± 0.0048** | 0.0339 ± 0.0358 | **0.4057 ± 0.0078** | 0.0579 ± 0.0286 | **0.3785 ± 0.0072** | 0.0704 ± 0.0348 |
| drug_target_GPCR_2009 | **0.9406 ± 0.0184** | 0.9315 ± 0.0235 | **0.9551 ± 0.0147** | 0.9525 ± 0.0178 | **0.8325 ± 0.0159** | 0.8151 ± 0.0192 | **0.8351 ± 0.0140** | 0.8212 ± 0.0154 |
| drug_target_HQ_2014 | **0.7307 ± 0.0139** | 0.7143 ± 0.0148 | **0.7863 ± 0.0145** | 0.7802 ± 0.0165 | **0.5658 ± 0.0142** | 0.5424 ± 0.0192 | **0.5805 ± 0.0117** | 0.5582 ± 0.0208 |
| drug_target_enzyme_2009 | **0.8447 ± 0.0206** | 0.8162 ± 0.0242 | **0.8852 ± 0.0168** | 0.8781 ± 0.0156 | **0.8123 ± 0.0185** | 0.8039 ± 0.0329 | **0.8265 ± 0.0156** | 0.8253 ± 0.0172 |
| drug_target_ionchannel_2009 | **0.9048 ± 0.0142** | 0.8724 ± 0.0408 | **0.9346 ± 0.0187** | 0.9144 ± 0.0171 | **0.8654 ± 0.0178** | 0.8325 ± 0.0186 | **0.8681 ± 0.0193** | 0.8418 ± 0.0235 |
| drug_target_moesm4_esm | **0.6851 ± 0.0094** | 0.6566 ± 0.0115 | **0.7391 ± 0.0055** | 0.7156 ± 0.0081 | **0.5766 ± 0.0089** | 0.5469 ± 0.0096 | **0.5922 ± 0.0080** | 0.5637 ± 0.0081 |
| industries_eductionfields_IPUMS | **0.3118 ± 0.0122** | 0.0887 ± 0.0620 | **0.4500 ± 0.0095** | 0.1290 ± 0.0700 | **0.4811 ± 0.0144** | 0.1470 ± 0.1136 | **0.4732 ± 0.0123** | 0.1580 ± 0.1151 |
| occupations_tasks_ONET | **0.4767 ± 0.0118** | 0.2187 ± 0.0168 | **0.6247 ± 0.0083** | 0.3234 ± 0.0098 | **0.5279 ± 0.0090** | 0.2336 ± 0.0195 | **0.5682 ± 0.0078** | 0.2742 ± 0.0158 |
| tfs_genes_regulation_ecoli | **0.5585 ± 0.0220** | 0.4886 ± 0.0329 | **0.6496 ± 0.0284** | 0.5820 ± 0.0259 | **0.4976 ± 0.0268** | 0.4563 ± 0.0305 | **0.5224 ± 0.0293** | 0.4680 ± 0.0315 |
| users_movies_movielens100k | **0.2200 ± 0.0040** | 0.1940 ± 0.0051 | **0.3384 ± 0.0083** | 0.2225 ± 0.1027 | **0.3548 ± 0.0065** | 0.3120 ± 0.0076 | **0.3588 ± 0.0061** | 0.3131 ± 0.0057 |
| export | **0.0521 ± 0.0030** | 0.0169 ± 0.0063 | **0.0837 ± 0.0041** | 0.0233 ± 0.0028 | **0.4596 ± 0.0111** | 0.1370 ± 0.0062 | **0.4190 ± 0.0068** | 0.1300 ± 0.0036 |
| amazon-product | **0.1246 ± 0.0022** | 0.1131 ± 0.0029 | **0.1796 ± 0.0020** | 0.1625 ± 0.0022 | **0.1154 ± 0.0020** | 0.1042 ± 0.0017 | **0.1321 ± 0.0018** | 0.1193 ± 0.0017 |

# N    THE EFFECTIVENESS OF CH THEORY OVER SIMPLE PARADIGM

In this work, we bridge the task of link prediction and recommendation using network science theory Cannistraci-Hebb theory, which captures the local-community structure inside of real-world networks and utilizes it for prediction. Instead of using existing trivial similarity measures, we introduce CH index to better build the mechanism and through the experimental results in the article, it shows that NSA shows stable and excellent ability in terms of recommendation performance. To better prove that NSA's take advantage of its intrinsic physic logic instead of the existing traditional scheme of collaborative filtering, we conducted ablation test, preserve the hyperparameter settings searched of NSA and replace the similarity measure from CH-index to CN (common neighbors, which is a basic similarity measure in the field of network science). Table 8 shows the results of with average results of 10 test splits on different datasets. It shows that NSA shows a clear advantage in terms of all the metrics. This could further prove that utilizing simple structure, NSA has its strong improvement over others with its root physic description of real-world networks.

# O    USAGE OF LLM

In this work, Large Language Model (LLM) is primarily used to assist with tasks such as text refinement, summarization, and improving the clarity and readability of the manuscript. The LLM helps streamline writing and editing, ensuring that technical content is clearly and accurately presented.

# P    NSA AGAINST NLGCL

In this section, we include new baseline NLGCL for comparison. For time reason, we tested on 10 middle scale networks. The test procedure remains the same as all the other methods, with 10 test-train splits and correspondingly 10 validation for hyperparameter choosing. The considered hyperparameters of NLGCL includes, learning rate in the range of [1e-3, 2e-3] and number of layers in the range of [2, 3]. Results are summarized in Table 9 with average results of 10 train-test splits reported. It shows that NSA is of advantage compared with this newest baseline, especially on datasets of higher sparsity, which further strengthen the effectiveness of NSA.

Table 9: **Results of NSA and NLGCL on different networks.** The bold value refers to the better performance.

| Dataset | Recall@10 | | Recall@20 | | NDCG@10 | | NDCG@20 | |
|---|---|---|---|---|---|---|---|---|
| | NSA | NLGCL | NSA | NLGCL | NSA | NLGCL | NSA | NLGCL |
| aidorganizations_issues | **79.55 ± 0.64** | 79.29 ± 0.56 | 95.79 ± 0.58 | **95.87 ± 0.31** | 59.68 ± 0.44 | **59.75 ± 0.40** | 65.04 ± 0.41 | **65.28 ± 0.31** |
| industries_eductionfields_IPUMS | 31.18 ± 0.39 | **32.67 ± 0.28** | 45.00 ± 0.30 | **47.65 ± 0.30** | 48.11 ± 0.45 | **50.73 ± 0.37** | 47.32 ± 0.39 | **50.25 ± 0.34** |
| congressmen_topics_US | **18.92 ± 0.12** | 18.62 ± 0.09 | **29.88 ± 0.15** | 28.82 ± 0.13 | **40.57 ± 0.25** | 39.68 ± 0.11 | **37.85 ± 0.23** | 37.11 ± 0.10 |
| drug_target_ionchannel_2009 | **90.48 ± 0.45** | 86.94 ± 0.99 | **93.46 ± 0.59** | 91.47 ± 0.87 | **86.54 ± 0.56** | 80.67 ± 0.84 | **86.81 ± 0.61** | 81.71 ± 0.83 |
| drug_target_GPCR_2009 | **94.06 ± 0.58** | 93.68 ± 0.75 | 95.51 ± 0.47 | **96.11 ± 0.73** | **83.25 ± 0.50** | 79.00 ± 0.99 | **83.51 ± 0.44** | 79.55 ± 0.99 |
| occupations_tasks_ONET | **47.67 ± 0.37** | 42.98 ± 0.33 | **62.47 ± 0.26** | 57.65 ± 0.25 | **52.79 ± 0.28** | 47.67 ± 0.25 | **56.82 ± 0.25** | 51.86 ± 0.24 |
| tfs_genes_regulation_ecoli | **55.85 ± 0.69** | 40.00 ± 0.89 | **64.96 ± 0.90** | 49.14 ± 0.87 | **49.76 ± 0.85** | 37.23 ± 0.84 | **52.24 ± 0.93** | 39.46 ± 0.80 |
| amazon-product | **12.46 ± 0.07** | 10.11 ± 0.06 | **17.96 ± 0.06** | 14.92 ± 0.07 | **11.54 ± 0.06** | 9.05 ± 0.05 | **13.21 ± 0.06** | 10.58 ± 0.04 |
| drug_target_enzyme_2009 | **84.47 ± 0.65** | 83.42 ± 0.86 | **88.52 ± 0.53** | 88.16 ± 0.56 | **81.23 ± 0.58** | 79.46 ± 0.86 | **82.65 ± 0.49** | 80.83 ± 0.79 |
| drug_target_HQ_2014 | **73.07 ± 0.44** | 72.96 ± 0.96 | 78.63 ± 0.46 | **80.31 ± 0.60** | **56.58 ± 0.45** | 52.29 ± 0.51 | **58.05 ± 0.37** | 54.07 ± 0.37 |

