# OpenReview forum: "Network Shape Automata: a brain network inspired collaborative filter for link prediction in bipartite complex networks and recommendation systems"
_ICLR.cc/2026/Conference — Submitted to ICLR 2026_

### Official Review · Reviewer_mqXY · 2025-10-17

**Soundness:** 1
**Presentation:** 2
**Contribution:** 1
**Rating:** 2
**Confidence:** 5

**Summary:**

This paper proposes Network Shape Automata (NSA), a memory-based collaborative filtering method that incorporates the Cannistraci-Hebb (CH) theory from network science into similarity computation for link prediction and recommendation in bipartite graphs. The authors conduct extensive evaluations on 13 medium-scale and 3 large-scale recommendation datasets.

**Strengths:**

- Introduces the relatively novel CH theory from network science to the recommender systems community.

**Weaknesses:**

- [Mandatory] SCCF was published in KBS 2023; it is neither advanced nor influential, and thus unconvincing as a research motivation. Moreover, its performance is similar to LightGCN and is significantly uncompetitive compared to recent CL-based methods.

- [Mandatory] Table 1 is not cited in the main text.

- [Mandatory] The most recent baselines used are from 2023. Please add 2024–2025 baselines, such as BIGCF [1], RecDCL [2], WeightGCL [3], and NLGCL [4].

- [Mandatory] Please update the related work; the discussion lacks the latest studies.

- [Mandatory] There is a lack of detailed introduction to CH3.1-L2.

- [Mandatory] There is a lack of theoretical analysis or intuitive interpretation of NSA itself. The physical meaning of CH index components (di_k, de_k) in the context of recommendation (e.g., what user/item behaviors they correspond to) is not clarified, making the method resemble a "black-box" similarity function and undermining its claimed "interpretability."

- [Mandatory] The time complexity analysis (Appendix J) indicates a complexity of O((U+I)^3) for heterogeneous sparse networks, which is prohibitive for large-scale recommendation. The paper fails to adequately discuss this potential bottleneck or propose effective mitigation strategies.


Refs:

[1] Exploring the individuality and collectivity of intents behind interactions for graph collaborative filtering, SIGIR 2024.

[2] RecDCL: Dual Contrastive Learning for Recommendation, WWW 2024.

[3] Squeeze and Excitation: A Weighted Graph Contrastive Learning for Collaborative Filtering, SIGIR 2025.

[4] NLGCL: Naturally Existing Neighbor Layers Graph Contrastive Learning for Recommendation, RecSys 2025.

**Questions:**

Please refer to Weaknesses. Btw, I have some optional questions:

- [Optional] Given the high time complexity in power-law networks, are there feasible approximation computations or sampling strategies to make NSA applicable to ultra-large-scale scenarios?

---

> ### Author Response · Authors · 2025-12-03
>
> **Reply to weakness1:**
>
> *“SCCF was published in KBS 2023; it is neither advanced nor influential, and thus unconvincing as a research motivation. Moreover, its performance is similar to LightGCN and is significantly uncompetitive compared to recent CL-based methods.”*
>
> We sincerely thank the reviewer for point out this concern. To address the reviewer’s concern, we include new baseline NLGCL[1] for comparison. For time reason, we tested on 10 middle scale networks. The test procedure remains the same as all the other methods, with 10 test-train splits and correspondingly 10 validation for hyperparameter choosing. The considered hyperparameters of NLGCL includes, learning rate in the range of [1e-3, 2e-3] and number of layers in the range of [2, 3]. Results are summarized in table below with average results of 10 train-test splits reported. It shows that **NSA is of advantage** compared with this newest baseline, **especially on datasets of higher sparsity**, which further strengthen the effectiveness of NSA.
>
> |Dataset|Recall@10||Recall@20||NDCG@10||NDCG@20||
> |-|-|-|-|-|-|-|-|-|
> ||NSA|NLGCL|NSA|NLGCL|NSA|NLGCL|NSA|NLGCL|
> |aidorganizations_issues|**79.55±0.64**|79.29±0.56|95.79±0.58|**95.87±0.31**|59.68±0.44|**59.75±0.40**|65.04±0.41|**65.28±0.31**|
> |industries_eductionfields_IPUMS|31.18±0.39|**32.67±0.28**|45.00±0.30|**47.65±0.30**|48.11±0.45|**50.73±0.37**|47.32±0.39|**50.25±0.34**|
> |congressmen_topics_US|**18.92±0.12**|18.62±0.09|**29.88±0.15**|28.82±0.13|**40.57±0.25**|39.68±0.11|**37.85±0.23**|37.11±0.10|
> |drug_target_ionchannel_2009|**90.48±0.45**|86.94±0.99|**93.46±0.59**|91.47±0.87|**86.54±0.56**|80.67±0.84|**86.81±0.61**|81.71±0.83|
> |drug_target_GPCR_2009|**94.06±0.58**|93.68±0.75|95.51±0.47|**96.11±0.73**|**83.25±0.50**|79.00±0.99|**83.51±0.44**|79.55±0.99|
> |occupations_tasks_ONET|**47.67±0.37**|42.98±0.33|**62.47±0.26**|57.65±0.25|**52.79±0.28**|47.67±0.25|**56.82±0.25**|51.86±0.24|
> |tfs_genes_regulation_ecoli|**55.85±0.69**|40.00±0.89|**64.96±0.90**|49.14±0.87|**49.76±0.85**|37.23±0.84|**52.24±0.93**|39.46±0.80|
> |amazon-product|**12.46±0.07**|10.11±0.06|**17.96±0.06**|14.92±0.07|**11.54±0.06**|9.05±0.05|**13.21±0.06**|10.58±0.04|
> |drug_target_enzyme_2009|**84.47±0.65**|83.42±0.86|**88.52±0.53**|88.16±0.56|**81.23±0.58**|79.46±0.86|**82.65±0.49**|80.83±0.79|
> |drug_target_HQ_2014|**73.07±0.44**|72.96±0.96|78.63±0.46|**80.31±0.60**|**56.58±0.45**|52.29±0.51|**58.05±0.37**|54.07±0.37|
>
> [1] NLGCL: Naturally Existing Neighbor Layers Graph Contrastive Learning for Recommendation, RecSys 2025.
>
>
> **Reply to weakness2:**
>
> *“Table 1 is not cited in the main text.”*
>
> We thank the reviewer for pointing this out. We have corrected this issue in the revised version and now properly cite Table 1 in the main text.
>
>
> **Reply to weakness3:**
>
> *“The most recent baselines used are from 2023. Please add 2024–2025 baselines, such as BIGCF [1], RecDCL [2], WeightGCL [3], and NLGCL [4].”*
>
> We thank the reviewer for the suggestion. Due to time limitations, we include only the latest baseline NLGCL, as mentioned in our reply to Weakness 1. The results clearly show that NSA performs favorably in most cases, especially on sparse networks.
>
>
> **Reply to weakness4:**
>
> *“Please update the related work; the discussion lacks the latest studies.”*
>
> We sincerely thank the reviewer for point this out. We’ve now updated the related work with CL-based method NLGCL discussed.
>
> **Reply to weakness5:**
>
> *“There is a lack of detailed introduction to CH3.1-L2.”*
>
> We thank the reviewer for pointing this out. We have revised the manuscript to provide a more detailed introduction to CH3.1-L2.
>
> **Reply to weakness6:**
>
> *“There is a lack of theoretical analysis or intuitive interpretation of NSA itself. The physical meaning of CH index components (di_k, de_k) in the context of recommendation (e.g., what user/item behaviors they correspond to) is not clarified, making the method resemble a "black-box" similarity function and undermining its claimed "interpretability."”*
>
> We sincerely thank the reviewer for raising this point. In CH theory, $di_k$ and $de_k$ represent, respectively, the number of internal connections within the local community and the number of external links. The local-community paradigm is defined for real-world networks, including recommendation-related social or user–item structures.

---

> ### Author Response · Authors · 2025-12-03
>
> **Reply to weakness7:**
>
> *“The time complexity analysis (Appendix J) indicates a complexity of O((U+I)^3) for heterogeneous sparse networks, which is prohibitive for large-scale recommendation. The paper fails to adequately discuss this potential bottleneck or propose effective mitigation strategies.”*
>
> We sincerely thank the reviewer about this. We apologize if we didn’t make it clear, we discussed about the possible scalability limitation of NSA in Appendix I. Also, we want to stress that middle-scale real-world recommendation usage should not be neglected. Our experiments are conducted on several real-world networks, including Drug-Target interaction, MovieLens-100K, Export-Country trade, and Amazon Product co-purchase datasets. While these networks may appear middle in scale, they represent industrially important applications with substantial real-world impact. For instance, the pharmaceutical industry, corresponding to the drug-target networks, has a global market size exceeding 1.5 trillion dollars; the entertainment and media sector, associated with the MovieLens-100K dataset, is projected to reach 3.5 trillion dollars by 2029; cross-border trade, reflected in the export-country network, accounts for over 32 trillion dollars annually. These examples illustrate that the networks we study, though not extremely large, are representative of high-value industrial systems, which should not be neglected. In many of these domains, network sizes remain moderate due to natural constraints, yet the demand for accurate, interpretable, and efficient recommendations is of value for these industries. Our method, NSA, is specifically fit for these practical scenarios. It is training-free, computationally efficient, and able to reuse key components across validation, making it fast and lightweight, while still delivering stable and competitive performance across all tested datasets. We hope this explanation underscores the real-world relevance and impact of our method also for industrial application such as the ones discussed above.
>
>
> **Reply to question1:**
>
> *”Given the high time complexity in power-law networks, are there feasible approximation computations or sampling strategies to make NSA applicable to ultra-large-scale scenarios?”*
>
> We thank the reviewer for this insightful question. Due to time constraints, we were unable to extend our discussion to approximation or sampling strategies. We agree that exploring such directions constitutes a promising avenue for future work.

---

### Official Review · Reviewer_yxjj · 2025-10-24

**Soundness:** 3
**Presentation:** 4
**Contribution:** 3
**Rating:** 6
**Confidence:** 4

**Summary:**

This paper proposes a novel memory-based collaborative filtering (CF) method called "Network Shape Automata" (NSA) for link prediction in bipartite networks. This work contrasts with the current mainstream trend of using deep learning and Graph Neural Networks (GNNs) for recommendation.

The core contribution of NSA is a novel, brain-inspired similarity metric derived from the Cannistraci-Hebb (CH) theory in network science. This metric (e.g., the CH3-L2 index) goes beyond simple common neighbors, instead evaluating the local community topology of node pairs to compute similarity.

The entire NSA method is a 4-stage, training-free process:

1. **CH Scoring:** Computes pairwise node similarity for users and items separately, using a CH index, a denominator for scaling, and an exponential parameter.
2. **Monopartite Projection:** Projects these similarities as weights onto user-user and item-item graphs.
3. **Bipartite Scoring:** Aggregates scores from the monopartite graphs to predict scores for unobserved user-item links.
4. **Mixing:** Linearly combines the user-based and item-based scores using a mixing parameter $\lambda$.

The authors conduct an extremely rigorous and comprehensive evaluation on 13 medium-sized bipartite networks from diverse domains (social and biological) and 3 large-scale recommendation datasets (Gowalla, Yelp2018, Amazon-book). The results show that the training-free NSA consistently matches or surpasses various state-of-the-art (SOTA) memory-based (SSCF) and model-based (e.g., LightGCN, BSPM, LT-OCF) baselines in performance.

The paper's main contribution is bridging network science theory with recommendation systems, providing a powerful, interpretable, and highly competitive non-learning baseline that challenges the necessity of using complex learned representations for this task.

**Strengths:**

1. Exceptional Empirical Performance: The method's primary strength is its performance. On 13 diverse datasets, NSA consistently achieves the 1st or 2nd average rank against 9 strong baselines (Figure 2) and remains highly competitive on the 3 large-scale benchmarks (Table 2).
2. Methodological Rigor: The experimental design is a model of good scientific practice. The use of 13+3 datasets, multiple realizations, and an extensive and fair hyperparameter search for *all* methods makes the results highly credible.
3. High Originality and Novel Perspective: This paper brings a genuinely new idea to the CF community by introducing the Cannistraci-Hebb (CH) theory from network science. This moves beyond standard heuristics and provides a brain-inspired, topologically-aware theoretical grounding for the similarity metric.
4. Interpretability and Simplicity: As a memory-based, training-free method, NSA is far simpler and more interpretable than its deep learning competitors. Its success demonstrates the immense predictive power of well-engineered topological features over end-to-end learned representations.
5. Robustness: The method shows strong performance in sparse network conditions. Furthermore, the ablation in Appendix H shows that even a simplified version (exponent parameter fixed to 1) is still one of the top-performing methods, highlighting the robustness of its core CH index.

**Weaknesses:**

1. Scalability Concerns: The most significant weakness (which the authors acknowledge) is computational complexity. The time complexity analysis in Appendix J shows $\mathcal{O}((U+I)^3)$ for sparse heterogeneous networks (dominated by CH scoring) or $\mathcal{O}(U^2I + UI^2)$ for homogeneous networks (dominated by CF aggregation). This is polynomially worse than GNNs at $\mathcal{O}(T \times E \times D)$. For truly industrial-scale graphs with billions of nodes, this approach may be infeasible. NSA is at a computational disadvantage compared to GNN methods like LightGCN which have lower complexity.
2. Limited Exploration of CH Theory: The paper only uses L2 (path length 2) indices (CH3-L2, CH3.1-L2). While these are sophisticated variants of common neighbors, the CH theory also includes multi-scale (Ln) indices. The paper does not explore L3 or higher-order paths. This feels like a missed opportunity, as it would offer a more direct point of comparison to multi-layer GNNs that aggregate multi-hop neighborhood information.
3. "Training-Free" vs. "Expensive Validation": The paper correctly claims the final model is "training-free." However, it relies on a massive hyperparameter search (claiming over 105,300 model evaluations) to find the optimal combination of CH index, denominator, exponent, scoring rule, and $\lambda$ mixer. This exhaustive validation search is a form of model selection, and its computational cost may be comparable to or *greater* than training one GNN. The paper claims it is "training-free" but relies on an extremely expensive, computationally intensive hyperparameter search space. The authors claim "over 105,300 model evaluations." This is not "simplicity"; it is a computationally expensive brute-force search. This form of "meta-training" might be computationally more expensive than training a GNN, making the "training-free" advantage almost moot. Is the grid search time factored into the time comparisons?
4. The ablation in Appendix N only compares the CH index against the most basic "Common Neighbors" (CN). This is a weak baseline. A comparison against more advanced, yet still simple, network science heuristics like Adamic-Adar (AA) or Resource Allocation (RA) would be more informative. A sophisticated variant of CH index and RA are similar, which is also worth considering.

**Questions:**

1. Practical Scalability: Given the polynomial complexity, how do the authors envision this method being practically applied to web-scale recommendation systems? The authors briefly mention "sampling strategies" in future work. Could they elaborate? For example, can the CH index be effectively approximated? Or could NSA be applied to sampled subgraphs while retaining its performance?
2. Higher-Order Paths (L3+): The CH-L2 indices are clearly a powerful enhancement over standard common neighbors. Did the authors experiment with or consider the L3 (path length 3) indices from CH theory? This seems like a natural extension to capture higher-order topological information and would be a good comparison point to a 3-layer GNN.
3. Cost of Validation Search: Can the authors provide a more direct comparison of the total wall-clock time required to run the *full hyperparameter validation search* for NSA (e.g., on the Yelp2018 dataset) versus the *total training and validation time* for a key baseline like LightGCN? Table 5 compares single runs, but the search space for NSA (index, denominator, exponent, rule, $\lambda$) seems very large. It is important to know the full cost of finding the "optimal" training-free model.

---

> ### Author Response · Authors · 2025-12-03
>
> **Reply to weakness1:**
>
> *“Scalability Concerns: The most significant weakness is computational complexity. The time complexity analysis in Appendix J shows  for sparse heterogeneous networks (dominated by CH scoring) or  for homogeneous networks (dominated by CF aggregation). This is polynomially worse than GNNs at . For truly industrial-scale graphs with billions of nodes, this approach may be infeasible. NSA is at a computational disadvantage compared to GNN methods like LightGCN which have lower complexity.”*
>
> Thanks for the reviewer’s concern. As we discussed in Appendix I, the scalability to extremely large networks might be an bottleneck of NSA. However, we would like to stress that middle-scale real-world recommendation usage should not be neglected. Our experiments are conducted on several real-world networks, including Drug-Target interaction, MovieLens-100K, Export-Country trade, and Amazon Product co-purchase datasets. While these networks may appear middle in scale, they represent industrially important applications with substantial real-world impact. For instance, the pharmaceutical industry, corresponding to the drug-target networks, has a global market size exceeding 1.5 trillion dollars; the entertainment and media sector, associated with the MovieLens-100K dataset, is projected to reach 3.5 trillion dollars by 2029; cross-border trade, reflected in the export-country network, accounts for over 32 trillion dollars annually. These examples illustrate that the networks we study, though not extremely large, are representative of high-value industrial systems, which should not be neglected. In many of these domains, network sizes remain moderate due to natural constraints, yet the demand for accurate, interpretable, and efficient recommendations is of value for these industries. Our method, NSA, is specifically fit for these practical scenarios. It is training-free, computationally efficient, and able to reuse key components across validation, making it fast and lightweight, while still delivering stable and competitive performance across all tested datasets. We hope this explanation underscores the real-world relevance and impact of our method also for industrial application such as the ones discussed above.
>
> **Reply to weakness2:**
>
> *“Limited Exploration of CH Theory: The paper only uses L2 (path length 2) indices (CH3-L2, CH3.1-L2). While these are sophisticated variants of common neighbors, the CH theory also includes multi-scale (Ln) indices. The paper does not explore L3 or higher-order paths. This feels like a missed opportunity, as it would offer a more direct point of comparison to multi-layer GNNs that aggregate multi-hop neighborhood information.”*
>
> We appreciate the reviewer’s insightful suggestion. Due to time constraints, we were unable to explore higher-order CH indices in this work. We also wish to clarify that in NSA, we first project the bipartite user–item graph into two monopartite graphs (user–user and item–item) and then apply L2 indices. This projection means that L2 on the projected graph corresponds to length-4 paths in the original bipartite graph, which already captures sufficiently rich multi-hop dependencies.
>
>
> **Reply to weakness 3:**
>
> *“"Training-Free" vs. "Expensive Validation": The paper correctly claims the final model is "training-free." However, it relies on a massive hyperparameter search (claiming over 105,300 model evaluations) to find the optimal combination of CH index, denominator, exponent, scoring rule, and  mixer. This exhaustive validation search is a form of model selection, and its computational cost may be comparable to or greater than training one GNN. The paper claims it is "training-free" but relies on an extremely expensive, computationally intensive hyperparameter search space. The authors claim "over 105,300 model evaluations." This is not "simplicity"; it is a computationally expensive brute-force search. This form of "meta-training" might be computationally more expensive than training a GNN, making the "training-free" advantage almost moot. Is the grid search time factored into the time comparisons?”*
>
> We thank the reviewer for this question. We would like to clarify that our hyperparameter validation is designed to ensure fair comparisons across all baselines, and does not undermine the training-free nature of NSA. Model-based methods typically involve continuous hyperparameters (learning rates, weight decay, dropout, number of layers, embedding size, etc.), making their search space significantly larger and often more computationally expensive than the discrete search space of NSA. Regarding time comparisons, as stated in the paper, the reported runtime corresponds to a single configuration for each method, ensuring fairness. Each baseline, whether model-based or training-free, requires hyperparameter selection, and NSA is not at a relative disadvantage, particularly given its lack of gradient-based training iterations.

---

> ### Author Response · Authors · 2025-12-03
>
> **Reply to weakness4:**
>
> *“The ablation in Appendix N only compares the CH index against the most basic "Common Neighbors" (CN). This is a weak baseline. A comparison against more advanced, yet still simple, network science heuristics like Adamic-Adar (AA) or Resource Allocation (RA) would be more informative. A sophisticated variant of CH index and RA are similar, which is also worth considering.”*
>
> We appreciate this helpful suggestion. Due to time constraints, we were unable to expand the ablation study to include other indices such as Adamic–Adar (AA) or Resource Allocation (RA). Our choice of CN was intentional, aiming to highlight the advantage of CH indices as a brain-inspired alternative to this classical heuristic. Exploring broader comparisons with other network-science indices is a valuable direction for future work.
>
>
> **Reply to question1:**
>
> *“Practical Scalability: Given the polynomial complexity, how do the authors envision this method being practically applied to web-scale recommendation systems? The authors briefly mention "sampling strategies" in future work. Could they elaborate? For example, can the CH index be effectively approximated? Or could NSA be applied to sampled subgraphs while retaining its performance?”*
>
> We thank the reviewer for this forward-looking question. Due to time constraints, we were unable to expand our discussion on potential sampling-based strategies or approximations for CH indices. We agree that exploring approximate CH computations or subgraph-based NSA variants is an important avenue for future research.
>
> **Reply to question2:**
>
> *“Higher-Order Paths (L3+): The CH-L2 indices are clearly a powerful enhancement over standard common neighbors. Did the authors experiment with or consider the L3 (path length 3) indices from CH theory? This seems like a natural extension to capture higher-order topological information and would be a good comparison point to a 3-layer GNN.”*
>
> We thank the reviewer for this suggestion. Due to time limitations we did not experiment with L3 or higher-order CH indices. As discussed in our reply to Weakness 2, the L2 index on the projected monopartite graph already corresponds to length-4 paths in the original bipartite graph, which is analogous to multi-hop information aggregation and thus not directly comparable to a 3-layer GNN.
>
> **Reply to question3:**
>
> *“Cost of Validation Search: Can the authors provide a more direct comparison of the total wall-clock time required to run the full hyperparameter validation search for NSA (e.g., on the Yelp2018 dataset) versus the total training and validation time for a key baseline like LightGCN? Table 5 compares single runs, but the search space for NSA (index, denominator, exponent, rule, ) seems very large. It is important to know the full cost of finding the "optimal" training-free model.”*
>
> We thank the reviewer for this question. Due to time constraints, we are unable to provide additional wall-clock comparisons beyond the single-run statistics included in the paper. As noted, the total search cost for all methods, including model-based baselines, must be multiplied by the number of hyperparameter settings tested. We emphasize again that the search space for model-based approaches is typically much larger (continuous hyperparameters) than that of NSA, so the validation cost should not be viewed as a disadvantage of NSA.

---

### Official Review · Reviewer_p9Mu · 2025-11-01

**Soundness:** 1
**Presentation:** 2
**Contribution:** 2
**Rating:** 2
**Confidence:** 5

**Summary:**

This paper proposes NSA, a memory-based collaborative filtering method that applies Cannistraci-Hebb theory from network science to bipartite recommendation networks. The authors show competitive performance on multiple datasets while maintaining interpretability and training-free inference.

**Strengths:**

1. The paper conducts an extensive empirical study on 16 datasets. It includes rigorous hyperparameter tuning with 10-fold validation for each realization, which strengthens the reliability of the results.

2. Integrating Cannistraci-Hebb theory from network science provides a principled and interpretable approach to similarity computation.

3. NSA requires no training phase and maintains full interpretability, which is valuable for deployment scenarios where model transparency and computational efficiency are important.

**Weaknesses:**

1. While the paper's core motivation emphasizes leveraging brain-inspired CH theory, the contribution appears closer to an engineering combination of existing components.
2. The SSCF and LT-OCF models used for comparison are from 2023 and 2021, respectively. Table 2 needs to include comparisons with more recent state-of-the-art baselines.
3. The paper presents only average rankings rather than full results across the 13 datasets used, making it difficult for readers to assess the actual performance contributions claimed by the authors.
4. For Amazon-Book, the evaluation uses an inconsistent methodology by testing only the CH3-L2 variant (NSA3) instead of the full NSA model.
5. Statistical significance testing appears necessary to adequately compare NSA against other baselines.
6. The term "automata" in the title is misleading. Network automata typically refer to dynamic systems with update rules, whereas NSA is a static similarity-based scoring function.
7. The notation is inconsistent. Section 3 uses U (users) and Γ (items) but later switches to U and I.

**Questions:**

1. Why does Table 5 show that the NSA runtime for amazon-product takes longer than BSPM?
2. Can you provide full results across all 13 datasets used, rather than just average rankings?
3. Given that SSCF and LT-OCF are models from 2023 and 2021, can you include comparisons with more recent baselines in Table 2?
4. Table 2 shows NSA3 (simplified NSA) used for Amazon-Book, but full NSA for Gowalla and Yelp2018. Why was the full NSA not tested on Amazon-Book?
5. What is the conceptual connection regarding CH theory between brain neural networks and user-item recommendation networks? Is this a superficial analogy rather than a deep theoretical connection?

---

> ### Author Response · Authors · 2025-12-03
>
> **Reply to weakness 1:**
>
> *“While the paper's core motivation emphasizes leveraging brain-inspired CH theory, the contribution appears closer to an engineering combination of existing components.”*
>
> We sincerely thank the reviewer for this comment. We would like to emphasize that our contributions go beyond an engineering-level combination of existing modules. The main contributions of this work are as follows:
> 1. We propose NSA, a memory-based collaborative filtering framework that incorporates CH theory into similarity computation using local network topology.
> 2. NSA is evaluated across 13 datasets covering both recommendation and link-prediction tasks. The method consistently demonstrates stable and often superior performance. Extensive hyperparameter tuning ensures fair and reproducible comparisons.
> 3. NSA achieves strong and stable performance on large-scale datasets such as Gowalla, Yelp2018, and Amazon-Book, demonstrating good scalability.
> 4. NSA effectively utilizes structural characteristics of networks to better capture meaningful user–item relationships. It maintains high accuracy even under sparse interactions while retaining interpretability.
>
> **Reply to weakness2:**
>
> *“The SSCF and LT-OCF models used for comparison are from 2023 and 2021, respectively. Table 2 needs to include comparisons with more recent state-of-the-art baselines.”*
>
> We sincerely thank the reviewer for point out this concern. To address the reviewer’s concern, we include new baseline NLGCL[1] for comparison. For time reason, we tested on 10 middle scale networks. The test procedure remains the same as all the other methods, with 10 test-train splits and correspondingly 10 validation for hyperparameter choosing. The considered hyperparameters of NLGCL includes, learning rate in the range of [1e-3, 2e-3] and number of layers in the range of [2, 3]. Results are summarized in table below with average results of 10 train-test splits reported. It shows that **NSA is of advantage** compared with this newest baseline, **especially on datasets of higher sparsity**, which further strengthen the effectiveness of NSA.
>
> |Dataset|Recall@10||Recall@20||NDCG@10||NDCG@20||
> |-|-|-|-|-|-|-|-|-|
> ||NSA|NLGCL|NSA|NLGCL|NSA|NLGCL|NSA|NLGCL|
> |aidorganizations_issues|**79.55±0.64**|79.29±0.56|95.79±0.58|**95.87±0.31**|59.68±0.44|**59.75±0.40**|65.04±0.41|**65.28±0.31**|
> |industries_eductionfields_IPUMS|31.18±0.39|**32.67±0.28**|45.00±0.30|**47.65±0.30**|48.11±0.45|**50.73±0.37**|47.32±0.39|**50.25±0.34**|
> |congressmen_topics_US|**18.92±0.12**|18.62±0.09|**29.88±0.15**|28.82±0.13|**40.57±0.25**|39.68±0.11|**37.85±0.23**|37.11±0.10|
> |drug_target_ionchannel_2009|**90.48±0.45**|86.94±0.99|**93.46±0.59**|91.47±0.87|**86.54±0.56**|80.67±0.84|**86.81±0.61**|81.71±0.83|
> |drug_target_GPCR_2009|**94.06±0.58**|93.68±0.75|95.51±0.47|**96.11±0.73**|**83.25±0.50**|79.00±0.99|**83.51±0.44**|79.55±0.99|
> |occupations_tasks_ONET|**47.67±0.37**|42.98±0.33|**62.47±0.26**|57.65±0.25|**52.79±0.28**|47.67±0.25|**56.82±0.25**|51.86±0.24|
> |tfs_genes_regulation_ecoli|**55.85±0.69**|40.00±0.89|**64.96±0.90**|49.14±0.87|**49.76±0.85**|37.23±0.84|**52.24±0.93**|39.46±0.80|
> |amazon-product|**12.46±0.07**|10.11±0.06|**17.96±0.06**|14.92±0.07|**11.54±0.06**|9.05±0.05|**13.21±0.06**|10.58±0.04|
> |drug_target_enzyme_2009|**84.47±0.65**|83.42±0.86|**88.52±0.53**|88.16±0.56|**81.23±0.58**|79.46±0.86|**82.65±0.49**|80.83±0.79|
> |drug_target_HQ_2014|**73.07±0.44**|72.96±0.96|78.63±0.46|**80.31±0.60**|**56.58±0.45**|52.29±0.51|**58.05±0.37**|54.07±0.37|
>
> [1] NLGCL: Naturally Existing Neighbor Layers Graph Contrastive Learning for Recommendation, RecSys 2025.
>
> **Reply to weakness 3:**
>
> *“The paper presents only average rankings rather than full results across the 13 datasets used, making it difficult for readers to assess the actual performance contributions claimed by the authors.”*
>
> We apologize if we didn’t make it clear. The individual results of each methods are reported in Appendix F and Appendix G for space limit, as what we comment in the paragraphs in the main text of results section line 362 and line 370.

---

> ### Author Response · Authors · 2025-12-03
>
> **Reply to weakness4:**
>
> *“For Amazon-Book, the evaluation uses an inconsistent methodology by testing only the CH3-L2 variant (NSA3) instead of the full NSA model.”*
>
> We thank the reviewer for this question. As explained in the results section (Line 377), NSA3.1 was omitted due to computational constraints. Based on the observations from the other large datasets (Gowalla and Yelp2018), CH3-L2 consistently outperforms CH3.1-L2. Therefore, excluding CH3.1-L2 does not adversely impact the conclusions, since it is only a hyperparameter choice and including it would likely improve the performance rather than reduce it.
>
> **Reply to weakness5:**
>
> *“Statistical significance testing appears necessary to adequately compare NSA against other baselines.”*
>
> We sincerely thank the reviewer for the question. We tested different baselines against NSA on 13 datasets, with 10 train-test split each. According to the average ranking and also the individual results, NSA outperforms other baselines which we think would be sufficiently to underscore the effectiveness of NSA over other baselines.
>
> **Reply to weakness6:**
>
> *“The term "automata" in the title is misleading. Network automata typically refer to dynamic systems with update rules, whereas NSA is a static similarity-based scoring function.”*
>
> We appreciate the reviewer’s comment. Here, “automata” is used in the sense that NSA adapts to and fits the structural characteristics (shapes) of networks with different topologies, rather than implying dynamical update rules in the classical automata sense.
>
> **Reply to weakness7:**
>
> *“The notation is inconsistent. Section 3 uses U (users) and Γ (items) but later switches to U and I.”*
>
> We thank the reviewer for pointing this out. We want to stress that in the main text Section 3, U and Γ refers to the set of users and items, while in Appendix J where we discuss about time complexity, U and I refers to the number of users and items. They are of different physical meanings.
>
> **Reply to question1:**
>
> *”Why does Table 5 show that the NSA runtime for amazon-product takes longer than BSPM?”*
>
> We sincerely thank the reviewer for pointing this out. As shown in Table 3 in Appendix B, Amazon-Product is the largest among the 13 medium-scale networks. As dataset size increases, the CPU-based computation in NSA becomes the bottleneck, especially compared to BSPM which is GPU-based and does not require iterative processing during training.
>
> **Reply to question2:**
>
> *“Can you provide full results across all 13 datasets used, rather than just average rankings?”*
>
> As noted in our response to Weakness 3, all detailed results are already reported in Appendix F and Appendix G, including both rankings and concrete metric values.
>
> **Reply to question3:**
>
> *”Given that SSCF and LT-OCF are models from 2023 and 2021, can you include comparisons with more recent baselines in Table 2?”*
>
> Please refer to our reply to Weakness 2. NSA outperforms the recent 2025 baseline NLGCL across multiple datasets. Due to time constraints, we were unable to perform experiments on additional large-scale datasets.
>
> **Reply to question4:**
>
> *“Table 2 shows NSA3 (simplified NSA) used for Amazon-Book, but full NSA for Gowalla and Yelp2018. Why was the full NSA not tested on Amazon-Book?”*
>
> Please refer to our reply to weakness4.
>
> **Reply to question5:**
>
> *“What is the conceptual connection regarding CH theory between brain neural networks and user-item recommendation networks? Is this a superficial analogy rather than a deep theoretical connection?”*
>
> We sincerely thank the reviewer for pointing this out. We want to stress that CH theory and local-community paradigm is inspired by the brain neural networks, while the it is proved to be effective in describing the dynamic changes of different kind of real-world networks, including the social networks or protein interactome networks [1]. We believe that the user-item network, as an instantiation of real-world social networks, would also be feasible to be described by this paradigm so that we tried to utilize it to promote the recommendation.
>
> [1] Cannistraci, C. V., Alanis-Lobato, G., & Ravasi, T. (2013). From link-prediction in brain connectomes and protein interactomes to the local-community-paradigm in complex networks. Scientific reports, 3(1), 1613.

---

### Meta-Review · Area_Chair_wepQ · 2026-01-07

**Summary:**

This paper proposes Network Shape Automata (NSA), a memory-based collaborative filtering method that applies Cannistraci-Hebb (CH) theory from network science to bipartite recommendation networks. The method is training-free and aims to bridge link prediction and recommendation tasks through brain-inspired network topology analysis.

While the paper presents an interesting cross-disciplinary approach and demonstrates competitive empirical results on medium-scale datasets, the outstanding concerns regarding scalability, theoretical depth, experimental consistency, and overstated claims (training-free, interpretability, automata terminology) outweigh the contributions. The rebuttal, while earnest, did not sufficiently address the core weaknesses identified by the reviewers.

**Reviewer Concerns:**

All three reviewers raised significant concerns about the paper's scalability limitations, with high complexity making the method impractical for large-scale recommendation systems. Reviewers consistently noted that the baselines used for comparison were outdated, and while the authors added NLGCL comparisons during rebuttal, this was limited to medium-scale datasets only. The claimed "training-free" advantage was questioned given the expensive hyperparameter search requiring over 105,000 model evaluations, and the authors did not provide requested wall-clock comparisons. Additionally, reviewers found the theoretical contribution lacking depth.

**Reviewer Scores:**

Reviewer p9Mu (Score: 2): Would likely maintain score. Core concerns about statistical significance, methodology inconsistency, and the superficial nature of the brain-network analogy were not convincingly addressed.

Reviewer yxjj (Score: 6): Would likely maintain score.

Reviewer mqXY (Score: 2): Would likely maintain score. The addition of NLGCL comparisons partially addresses baseline concerns, but fundamental issues regarding theoretical depth, scalability, and interpretability remain unresolved.

---

### Decision · Program_Chairs · 2026-01-26

Reject